# Pushing detectability and sensitivity for subtle force to new limits with shrinkable nanochannel structured aerogel

Xinlei Shi[1], Xiangqian Fan[1], Yinbo Zhu[2], Yang Liu[1], Peiqi Wu[1], Renhui Jiang[3], Bao Wu[2], Heng-An Wu[2], He Zheng [3], Jianbo Wang [3], Xinyi Ji [1], Yongsheng Chen [4✉] & Jiajie Liang [1,4,5✉]

There is an urgent need for developing electromechanical sensor with both ultralow detection limits and ultrahigh sensitivity to promote the progress of intelligent technology. Here we propose a strategy for fabricating a soft polysiloxane crosslinked MXene aerogel with multilevel nanochannels inside its cellular walls for ultrasensitive pressure detection. The easily shrinkable nanochannels and optimized material synergism endow the piezoresistive aerogel with an ultralow Young's modulus (140 Pa), numerous variable conductive pathways, and mechanical robustness. This aerogel can detect extremely subtle pressure signals of 0.0063 Pa, deliver a high pressure sensitivity over 1900 kPa$^{-1}$, and exhibit extraordinarily sensing robustness. These sensing properties make the MXene aerogel feasible for monitoring ultra-weak force signals arising from a human's deep-lying internal jugular venous pulses in a non-invasive manner, detecting the dynamic impacts associated with the landing and take-off of a mosquito, and performing static pressure mapping of a hair.

[1] School of Materials Science and Engineering, National Institute for Advanced Materials, Nankai University, 300350 Tianjin, China. [2] CAS Key Laboratory of Mechanical Behavior and Design of Materials, Department of Modern Mechanics, CAS Center for Excellence in Complex System Mechanics, University of Science and Technology of China, 230027 Hefei, China. [3] School of Physics and Technology, Center for Electron Microscopy, MOE Key Laboratory of Artificial Micro- and Nano-structures, and Institute for Advanced Studies, Wuhan University, 430072 Wuhan, China. [4] Key Laboratory of Functional Polymer Materials of Ministry of Education, College of Chemistry, Nankai University, 300350 Tianjin, China. [5] Tianjin Key Laboratory of Metal and Molecule-Based Material Chemistry and Collaborative Innovation Center of Chemical Science and Engineering (Tianjin), Nankai University, 300350 Tianjin, China. ✉email: yschen99@nankai.edu.cn; liang0909@nankai.edu.cn

The real-time sensing of subtle mechanical force signals is a key requirement for many next-generation cutting-edge intelligent applications[1,2]. There is an urgent demand for pressure sensors that can monitor extremely weak blood pressure waveforms (millipascal-scale) from deeply embedded veins in the human body, detect faint acoustic wave transmission in the air (tens of decibel scale), and recognize tiny signals induced by small movements or ultralight weights (milligram-scale) in intelligent healthcare, soft robot, and human–machine interface applications[3–5]. In addition, to enable the accurate sensing of subtle force signals, highly sensitive pressure sensors are needed because a low sensitivity often severely deteriorates the sensing resolution and leads to poor noise resistance, which limit the applications of such sensors in readout electronics[6,7]. Thus, the development of pressure sensors that combine an ultralow minimum detectable pressure limit with an ultrahigh sensitivity will offer exciting new opportunities for sensing applications, but significant materials-related challenges must be overcome.

Piezoresistive pressure sensors that rely on electrical resistance changes induced by material deformation in response to an applied pressure are currently the most promising force-sensing technology due to the broad selection of active materials and the simple fabrication and integration processes[7,8]. In principle, electrical resistance changes are governed by the structure and electrical features of piezoresistive materials, while material deformation critically depends on their geometry and mechanical properties[7,9]. Over the past few years, extensive investigations have focused on improving the detectability and/or sensitivity of piezoresistive sensors by optimizing the structural and electrical properties of sensing materials[8,10–14]. While conventional piezoresistive sensors made of bulk composites of conductive fillers and insulating polymers exhibit a low sensitivity[15], recent studies have demonstrated that hierarchical nano/microstructured materials (e.g., cellular monolith/sponges[8,16,17], nanomesh[18], microdomes/micropillars/microspines[10,11,19], bristled nanoparticles[12,20], and microchannels/multilayers[21,22]) provide decent sensing performance (sensitivity >100 kPa$^{-1}$). This is because when a piezoresistive material is compressed, more conductive pathways can be created in hierarchical nano/microstructures than in bulk structures, resulting in greater changes to the contact and/or internal resistance in hierarchically structured sensing materials[19,21,22]; however, the minimum pressure detection limit for most of these devices is still low (~1 Pa) due to the relatively large modulus of the rubbery substrates or soft matrixes[23]. Moreover, most of these hierarchically structured sensing materials are made of stiff conductive nanomaterials such as graphene, carbon nanotubes, MXenes, or nanostructured conducting polymers[8,9,24–27]. Such conductive materials usually possess intrinsically high Young's moduli and tend to show strong resistance to elastic deformation, further lowering their ability to detect subtle pressures[28].

Because the sensitivity of piezoresistive pressure sensors is generally inversely proportional to their Young's modulus[9], conductive aerogels have recently been investigated as an ideal alternative piezoresistive material for sensing subtle pressure changes due to their low modulus (or low mass density)[9,25,26]. Mathematically, the critical stress value ($\sigma_c \propto (t/l)^3$) that triggers the buckling or bending of cellular walls in aerogels is determined by the thickness ($t$) and length ($l$) of the cellular walls[27]. Soft conductive aerogels can be obtained by decreasing the cross-linking (or junction) density and mass density and increasing the aspect ratio ($l/t$) of cellular walls in the microcells. Previous work has reported that a minimum detectable pressure limit of 0.082 Pa was achieved using an elastic graphene aerogel with an ultralow density of 0.54 mg/cm$^3$ and a large aspect ratio in the cellular walls[25]. Nevertheless, this ultralight graphene aerogel delivered a low sensitivity of ~10 kPa$^{-1}$ due to a significant reduction in its

conductive pathway density. To retain their mechanical robustness and elasticity, insulating components are often compounded into conductive aerogels, which further diminishes their subtle pressure detectability and sensitivity[16,17,28,29]. Overall, although intense recent research efforts have advanced the development of piezoresistive materials for sensing applications[12,14,25,30], the simultaneous achievement of both an ultralow detection limit and ultrahigh sensitivity in piezoresistive materials is still challenging[10,11].

Herein, we propose a strategy to design a piezoresistive sensor for ultrasensitive subtle pressure sensing using a conductive sensing material based on an elastic and ultrasoft MXene-based aerogel with a hierarchical multilevel cellular wall structure. The core innovation of this design—the easily shrinkable multilevel nanochannels inside the cellular walls of the aerogel—was achieved by intercalating ultrasoft bottlebrush polysiloxane into MXene interlayers via covalent crosslinking. This hierarchical structure, together with the low-density structure of the MXene aerogel, endowed the piezoresistive material with an ultralow Young's modulus (~140 Pa at a density of 10 mg/cm$^3$), which significantly reduced the critical stress value that triggers material deformation. The shrinking of the multilevel nanochannels under compression increased the contact area between neighboring MXene nanosheets, which led to the formation of numerous new conductive paths and a considerable resistance change[21,22,31]. The formation of covalent crosslinks between the MXene and polysiloxane afforded the hierarchical aerogel with excellent mechanical robustness and compressive elasticity (up to 80%). Due to this sophisticated design, the resulting piezoresistive bottlebrush polysiloxane and MXene based aerogel could rapidly respond to an extremely low pressure of 0.0063 Pa with an ultrahigh sensitivity (>1900 kPa$^{-1}$). Moreover, this subtle pressure sensing performance, including the sensitivity and detectability, was retained even after 10,000 compress–release cycles. This piezoresistive aerogel can be assembled into flexible pressure sensing arrays for diverse cutting-edge applications that require the ability to detect extremely weak mechanical signals.

## Results

**Material design and sensing mechanism.** The design principle of our piezoresistive material involved combining the unique sensing performance of hierarchical nanostructures with an aerogel structure. First, we synthesized a piezoresistive aerogel from pure MXene nanosheets (abbreviated as MX-AG, where MX and AG represent MXene and aerogel, respectively) through the assembly and restacking of Ti$_3$C$_2$T$_x$ nanosheets via weak van der Waals interactions. Flexible Ti$_3$C$_2$T$_x$ MXene nanosheets were chosen as the conductive building block due to their high electrical conductivity, exceptional mechanical properties, and abundant reactive surface groups (i.e., Ti–OH)[22,32]. The piezoresistive sensing mechanism of MX-AG was dominated by external resistance changes ($R_E$) in the cellular walls induced by the bending or bucking of the cellular walls in the aerogel under pressure (Supplementary Fig. 1)[9]. A typical approach to realize an ultralow detection limit in MX-AG is to decrease the mass density and increase the aspect ratio ($l/t$) of the cellular walls in the aerogel (Supplementary Note 1)[27]; however, this method decreases the electrical and mechanical performance[27,33]. In contrast to this conventional strategy, our strategy uses Ti$_3$C$_2$T$_x$ MXene nanosheet and 3-glycidyloxypropyldimethoxymethylsilane (GPDMS) as the precursors to form composites with a variety of functional properties. GPDMS was selected to compound with MXene nanosheets based on three considerations. First, GPDMS can be hydrolyzed and polymerized into bottlebrush-like poly(3-

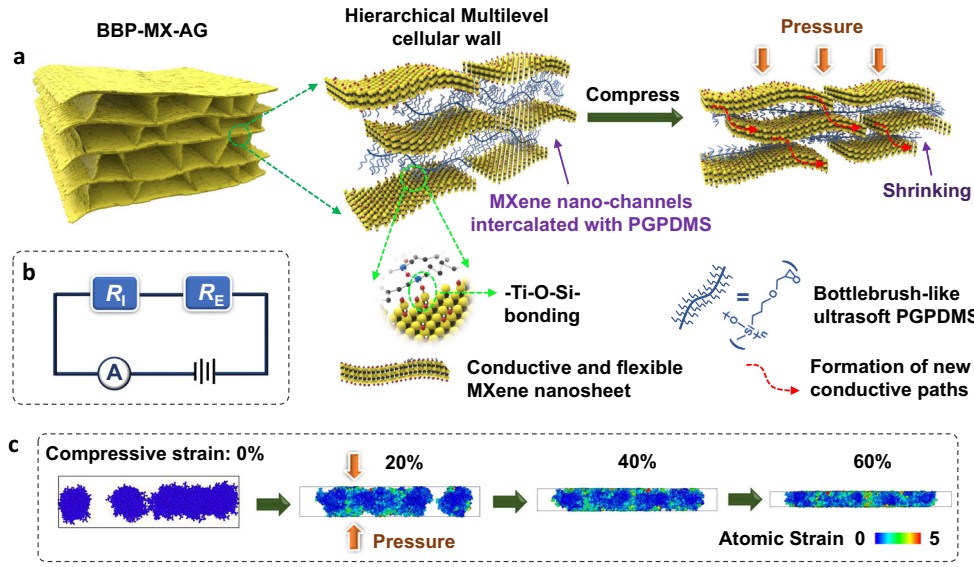

**Fig. 1 Schematic illustration of piezoresistive BBP–MX–AG. a** Illustration of the structure of BBP–MX–AG and the shrinking process of the multilevel cellular wall with bottlebrush-like PGPDMS crosslinked MXene nanochannels under pressure. **b** Equivalent circuit diagram of the piezoresistive BBP–MX–AG sensor. $R_I$ is the resistance change caused by shrinkage of the nanochannels inside the cellular walls, and $R_E$ is the resistance change induced by the bending and bucking of the cellular walls. **c** MD simulations demonstrating the atomic configurations of PGPDMS between the MXene nanochannels under compressive strain. The atoms are colored based on their atomic strain.

glycidoxypropyldimethoxymethylsilane) (PGPDMS) with a soft polysiloxane main chain and flexible short side-chains (Fig. 1a, Supplementary Figs. 2 and 3, and Supplementary Note 4). Second, bottlebrush polysiloxane-based materials are ultrasoft materials with moduli on the order of 100 Pa[33,34]. Third, the two ends of the main chain of PGPDMS can covalently bond with MXene nanosheets via the formation of Ti–O–Si bonds[35–38]. The MXene aerogel intercalated with bottlebrush PGPDMS (abbreviated as BBP–MX–AG, where BBP, MX, and AG represent the bottlebrush PGPDMS, MXene, and aerogel, respectively) can be obtained through a simple fabrication process consisting of a hydrothermal reaction, directional freeze-drying, and mild thermal annealing (described in detail in the experimental section). PGPDMS can act as a spacer for intercalation into the MXene interlayer, which prevents the restacking of MXene nanosheets[21,36,39–41]. This can facilitate the formation of easily shrinkable nanochannels inside the cellular walls (Fig. 1a)[21]. Such a hierarchical multilevel structure allows the detection of external forces by shrinking (or expanding) the spaces between nanochannels, which, in turn, leads to considerable resistance changes inside the cellular walls (denoted as an $R_I$ change)[21,22,31]. The total resistance changes ($R_{Total}$) of BBP–MX–AG in response to external forces have two primary contributions: $R_E$ and $R_I$ (Fig. 1b). In addition, the chemical stability (Supplementary Fig. 4), mechanical elasticity, and robustness of the nanochannels and the final aerogel geometry can be greatly enhanced by crosslinking the MXene with PGPDMS.

The sensing properties of piezoresistive aerogels can be tuned by controlling the synthesis conditions. For simplicity, unless otherwise noted, we focus our study on MXene-based aerogels with a mass density of ~10 mg/cm³ and with an initial MXene-to-silane weight ratio of 5:1 in the precursor mixtures. For comparison, we also fabricated MXene aerogels intercalated with an interchain-crosslinked poly(3-glycidoxypropyltrimethoxysilane) (PGPTMS) network, which is abbreviated as ICP-MX-AG (ICP represents interchain-crosslinked PGPTMS) (Supplementary Figs. 5 and 6). The shrinkage characteristics of the

nanochannels in the cellular walls of both BBP–MX–AG and ICP-MX–AG were investigated using molecular dynamics (MD) simulations (Supplementary Fig. 7 and Supplementary Note 2). Under pressure stimuli, as shown in the snapshots (Fig. 1c and Supplementary Fig. 8), the branched short chains of PGPDMS were compressed and underwent deformation first, and the branched chains had a much higher atomic strain. As the pressure increased further, the main chains of PGPDMS began to be curved and compressed freely due to their linear structure and elasticity. Moreover, the simulation results showed that the linear PGPDMS chains could be compressed more easily than the crosslinked PGPTMS networks (Supplementary Fig. 9).

**Structural and mechanical characterization.** We first characterized the morphology and structure of BBP–MX–AG using scanning electron microscopy (SEM), high-resolution transmission electron microscopy (HRTEM), and X-ray diffraction (XRD). Figure 2a displays a representative SEM image of BBP–MX–AG, which exhibits a honeycomb-like cellular structure. The cellular wall ($t = 12.5$ nm) in the BBP–MX–AG was made of multilayer MXene with an average nanochannel spacing ($d_A$) of 1.8 nm, as confirmed by HRTEM (Fig. 2b) and XRD (Fig. 2d). The nanochannel spacing of the cellular wall in ICP-MX–AG was also demonstrated to be about 1.8 nm (Supplementary Fig. 10). Energy-dispersive X-ray spectroscopy (EDS) element mapping illustrated the homogenous distribution of Si atoms (from PGPDMS) over the entire BBP–MX–AG scaffold (Supplementary Fig. 11). By contrast, the interlayer distance of the cellular walls in MX-AG was only 1.1 nm (Fig. 2c, d), which is consistent with the typical interlayer distance of pristine $Ti_3C_2T_x$ films. This indicates the close restacked structure of the cellular walls with no interlayer spacing (or intercalants) in MX–AG[22]. This confirms the successful intercalation of PGPDMS and PGPTMS into the MXene interlayer at the molecular level to form parallel nanochannel structures inside the cellular walls of BBP–MX–AG and ICP-MX–AG, respectively[36,39–41]. The Fourier-transform infrared (FT-IR) spectra (Supplementary Figs. 12 and 13) and X-ray photoelectron spectra (XPS, Supplementary Fig. 14) reveal that PGPDMS and PGPTMS covalently

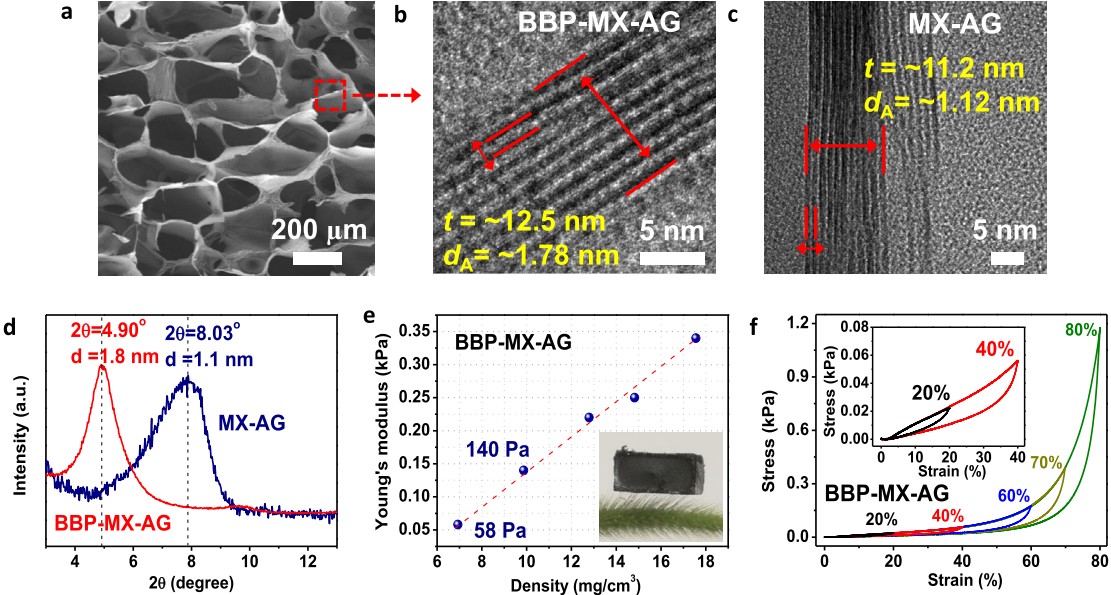

**Fig. 2 Structural and mechanical properties of BBP–MX–AG. a** SEM image of BBP–MX–AG. HRTEM images showing nanochannels with an interlayer spacing of (**b**) 1.8 nm in the cellular wall of BBP–MX–AG and **c** 1.1 nm in the cellular wall of MX–AG. **d** XRD patterns of BBP–MX–AG and MX–AG. **e** Young's modulus versus density of BBP–MX–AG. Inset: optical image showing a light monolithic sample sitting on top of a green bristlegrass. **f** Compressive stress–strain curves of BBP–MX–AG during compression-release cycles with maximum strains up to 80%.

crosslinked with the MXene nanosheets via the formation of Ti–O–Si bonds (Supplementary Note 3)[36–38,42]. These characterization results verify the hierarchical nanostructure of PGPDMS and PGPTMS intercalated multilevel cellular walls in the BBP–MX–AG and ICP–MX–AG, as illustrated in Fig. 1a.

Next, we investigated the mechanical properties of BBP–MX–AG through quasi-static uniaxial compress–release tests. Calculations using the linear elastic region of the stress–strain curves (Supplementary Fig. 15) showed that the compressive modulus of a light BBP–MX–AG sample (Fig. 2e, inset) achieved values as low as 58 and 140 Pa for samples with mass densities of approximately 7.0 and 10 mg/cm$^3$, respectively. These values indicate their ultrasoft characteristics. These Young's modulus values are more than two times lower than that of MX–AG (Supplementary Fig. 16), more than two orders of magnitude lower than those of most reported porous graphene, carbon nanotube, MXene, and polymer aerogels with similar densities[32,43,44], and approximately 1000 times lower than those of the conventional elastomers[33,45]. Such a low modulus enables the aerogel to deform under stimulation by a subtle pressure, enabling it to respond to weak force signals. Moreover, the monolithic BBP–MX–AG sample demonstrated a high recoverability during a uniaxial compress–release cycle, and a low residual strain (<1%) was observed after the first compression–release cycle with a maximum strain of 80% (Fig. 2f). The aerogel's height and ultimate stress remained at 96% and 83% of their original values, respectively, after 100 cycles at maximum compressive strains between 0% and 80% (Supplementary Fig. 17). Taken together, these results show that BBP–MX–AG exhibits attractive mechanical elasticity, robustness, and softness.

Furthermore, the shrinkage nature of the nanochannels in the cellular walls of BBP–MX–AG was characterized based using dynamic in-situ HRTEM observations. Figure 3 and Supplementary Fig. 18 demonstrated one shrinking–expansion cycle of the nanochannels. As shown in Fig. 3a–d, the thickness ($t$) of a cellular wall containing 5 MXene layers (indicated by the red line) decreased from 7.1 nm (1 s) to 5.01 nm (19 s) when an external force was applied, and then recovered to 6.87 nm (28 s) when the stimulating force was released. This cellular wall thickness change was induced by a simultaneous spacing change in the nanochannels of the cellular

walls, as confirmed by digital micrograph images (Fig. 3e–h). Specifically, the average distance ($d_A$) between two MXene layers decreased from an initial value of 1.78–1.6 nm and 1.25 nm at 15 and 19 s sequentially during the loading process, which then increased back to 1.72 nm at 28 s when unloaded (Supplementary Movie 1). This shrinkable nature of the multilevel cellular walls in BBP–MX–AG was also characterized by XRD measurements. As can be observed from the XRD patterns in Supplementary Fig. S19, the interlayer spacing of the nanochannels in the cellular walls of BBP–MX–AG decreased from an original value of 1.8 nm (without an external force) to 1.5 and 1.4 nm when external pressures of 0.2 and 1 Pa were applied to the sample, respectively. The distance reduction between two adjacent MXene layers in the cellular walls increased the number of nanosheet contact points and formed additional conduction pathways, resulting in a concurrent strong reduction in the electrical resistance[21,22,31].

**Subtle pressure sensing performance**. Subsequently, we measured the pressure-dependent electrical signal changes of the piezoresistive aerogel sensor under an ultralow pressure to evaluate the sensing behavior of piezoresistive BBP–MX–AG. The pressure sensor was constructed by integrating a monolithic aerogel on top of a flexible interdigital electrode-coated PET substrate (Supplementary Fig. 20). The dimensions of the resulting square pressure-sensitive pad were 0.5 cm × 0.5 cm. Subtle pressure stimuli were first applied by a mechanical testing stage equipped with a force sensor with an ultrasensitive force gauge (Supplementary Fig. 21a). Fig. 4a, b illustrate the time-resolved relative current changes of a BBP–MX–AG sensor under dynamic pressures (vibrational frequency of 1 Hz) and gradient static pressure stimuli. The sensing device could detect a pressure signal of 0.025 Pa and could perceive a pressure gradient of 0.025 Pa with stable current signal changes and a high signal/noise ratio, under both consecutive static and dynamic force stimuli. The response and relaxation times were both shorter than 50 ms during the application and release of pressure (Fig. 4d and Supplementary Fig. 22). Considering the contact time between the force sensor and our sensing device during tests, the real response and relaxation times may be even shorter.

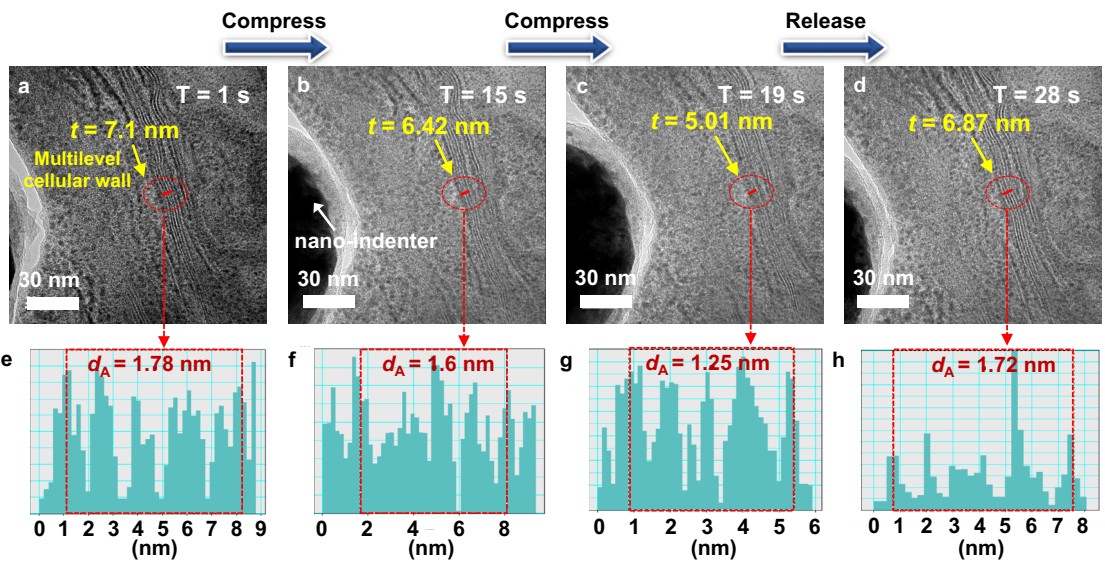

**Fig. 3 In-situ characterization of the shrinking and expansion of nanochannels in the cellular walls of BBP–MX–AG. a–d** In-situ cross-sectional HRTEM images showing the thickness changes in the cellular walls containing five MXene layers (labeled by red lines) caused by the shrinking and expansion of nanochannels during one compression-release cycle under an external force supplied by a nanoindenter. **e–h** Analysis of the specific spacing changes in the nanochannels of cellular walls in (**a**)–(**d**) using digital micrograph software.

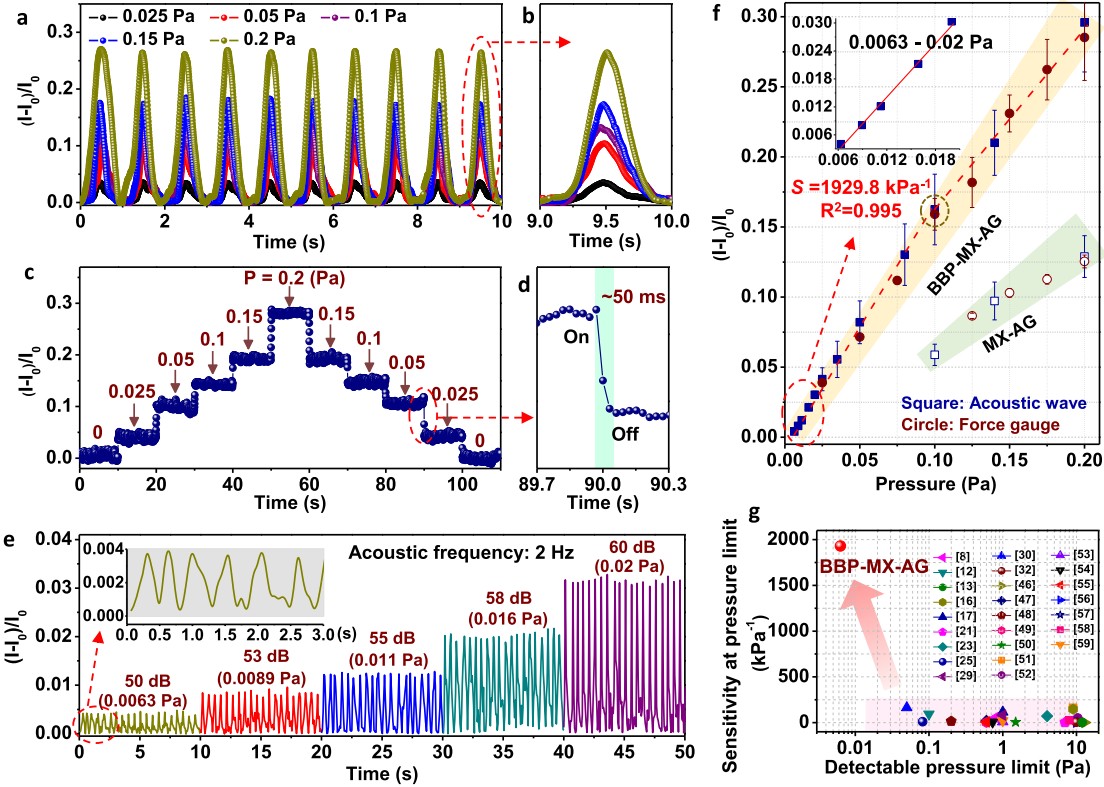

**Fig. 4 Characterization of subtle pressure sensing performance of the BBP–MX–AG sensor. a** The relative current changes over time of a BBP–MX–AG sensing device by varying the pressure with a vibrational frequency of 1 Hz in the pressure range of 0.025–0.2 Pa. **b** Magnified reversible compression–release behavior under different pressures. **c** Relative current changes over time of a BBP–MX–AG sensor to gradient static pressure in the pressure range of 0.025–0.2 Pa. **d** Magnified curves of the transient sensing response time to an applied pressure. **e** Relative current change over time of a sensing device due to a gradient acoustic wave with a sound intensity increasing from 50 to 60 dB (corresponding to sound pressures of 0.0063–0.02 Pa) under a fixed frequency of 2 Hz. **f** The subtle pressure sensitivity of the BBP–MX–AG (solid symbol) and pure MX–AG (hollow symbol) sensors stimulated by different pressure sources. Square and circle symbols represent the pressure sensors stimulated by acoustic waves and a force gauge, respectively. **g** Comparison of the minimum detectable pressure and corresponding sensitivity at a minimum pressure of the BBP–MX–AG piezoresistive sensor and previously reported state-of-the-art pressure sensors.

Notably, the sensing curve of the BBP–MX–AG sensor exhibited a steep slope under pressure stimuli <0.1 Pa (Fig. 4f). A clear decrease in the slope was observed for the sensing curve when the applied pressure was >0.1 Pa (Fig. 4f). The critical stress value that triggered the bending/bucking of cellular walls was calculated to be 0.1 Pa for BBP–MX–AG (Supplementary Note 1 and Supplementary Fig. 23). The resistance changes of BBP–MX–AG triggered by pressure stimuli <0.1 Pa do not rely on the bending or buckling of cellular walls ($R_E$). Instead, these changes are dominated by the shrinking of the nanochannels inside the cellular walls ($R_I$). Significantly, the pressure sensitivity in the pressure range of 0.025–0.1 Pa was calculated to be 1476.7 kPa$^{-1}$ with good linearity ($R^2 = 0.996$, Supplementary Fig. 24), which far exceeds the highest pressure sensitivity of most previously reported piezoresistive sensors[8,12,13,16,17,21,23,25,29,30,32,46–59]. When the applied pressure further entering into the pressure range over 150 Pa, corresponding to an abrupt stress increasing regime as can be seen in the compressive stress–strain curves in Fig. 2f, continuously decreasing in pore volume, nearby wall-to-wall contraction and subsequent densification of cells are happened in BBP–MX–AG, which majorly contributed to sensing mechanism in this pressure range[27,60,61].

It should be noted that 0.025 Pa is the minimum pressure that the mechanical testing system used in this work can supply. To determine the minimum detectable pressure limit of our BBP–MX–AG sensor, acoustic waves were employed as another external pressure stimulus that could be rapidly transmitted to and directly interacted with the aerogel sensor without a barrier (Supplementary Fig. 21b)[10,30]. Figure 4e and Supplementary Fig. 25 present the stable and well-defined current signal responses of the BBP–MX–AG sensor to acoustic waves generated from a sound source with different sound intensities at a fixed frequency of 2 Hz. The relative current changes increased as the sound intensities increased from 50 to 60 dB, and the output signal frequencies were consistent with the frequencies of the acoustic waves (Fig. 4e, inset). The acoustic intensity can be converted to sound pressure, and the corresponding sensing

curves measured by either a force gauge or acoustic waves exhibited a similar slope and sensitivity (Fig. 4f). It is important to note that the detectable sound level of 50 dB for our BBP–MX–AG sensor corresponds to a detectable pressure limit of approximately 0.0063 Pa. The acoustic pressure sensitivity in the sound intensity range of 50–60 dB (corresponding to a pressure range of 0.0063–0.02 Pa) can reach 1929.8 kPa$^{-1}$ (Fig. 4f), together with a detection limit of 0.0063 Pa, which significantly outperforms most reported pressure sensors (Fig. 4g and Supplementary Table 1)[8,12,13,16,17,21,23,25,29,30,32,46–59].

By contrast, the MX–AG sensor does not show an electrical response to pressure stimuli below 0.1 Pa, neither for those supplied by the mechanical testing stage nor for those supplied by the sound source (Supplementary Fig. 26). This indicates a minimum detectable pressure limit of approximately 0.1 Pa for this pure MXene aerogel without shrinkable nanochannel structures in its cellular walls. This value is largely consistent with the calculated $\sigma_c$ (Supplementary Note 1), revealing that the sensing behavior of MX–AG sensor was dominated by the bending or buckling of the cellular walls (corresponding to $R_E$) in the aerogel. Moreover, the detection limit of the pressure sensors assembled from ICP–MX–AG was only 0.1 Pa (Supplementary Fig. 27), despite the presence of shrinkable nanochannels in its cellular walls. MD simulations revealed that the crosslinked PGPTMS network exhibited a much higher compressive modulus than that of bottlebrush-like PGPDMS (Supplementary Fig. 9). Thus, a larger force stimulus was required to trigger the deformation of the crosslinked network of PGPTMS in the nanochannels. These results confirm that the intercalation of a soft polymer as the molecular spacer into the nanochannels of multilevel cellular walls plays a critical role in the fabrication of highly sensitive piezoresistive aerogels.

Since the sensing performance is strongly related to the mass density of a piezoresistive aerogel[9], the pressure sensitivity and detectability of BBP–MX–AG (density = ~7 mg/cm$^2$ and Young's modulus = 58 Pa; Fig. 2e) were also evaluated. Piezoresistive

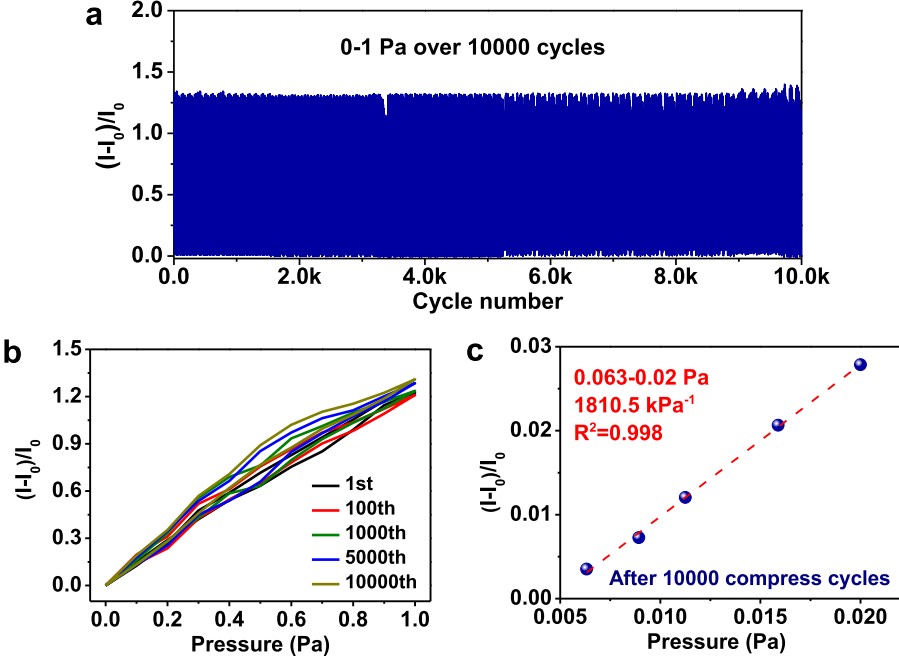

**Fig. 5 Subtle pressure sensing stability and durability of BBP–MX–AG. a** Relative current changes of the BBP–MX–AG sensor over 10,000 compression–release cycles in the 0–1 Pa range. **b** Detailed relative current changes versus pressure curves at the 1st, 100th, 1000th, 5000th, and 10,000th compression–release cycles corresponding to panel (**b**). **c** Relative current changes of the BBP–MX–AG sensor under pressures in the 0.0063–0.02 Pa range after undergoing 10,000 compression–release cycles in the 0–1 Pa range.

BBP–MX–AG (density = ~7 mg/cm$^2$) exhibited nearly the same pressure detection limit (0.0063 Pa) as the one with a density of ~10 mg/cm$^2$ (Supplementary Fig. 28). This is because the pressure detection limit is mainly determined by the shrinkable nanochannels in the cellular walls of BBP–MX–AG. Both BBP–MX–AG samples have similar multilevel nanochannels in their cellular walls because their initial MXene-to-silane weight ratio and fabrication conditions were the same; however, the sensitivity of BBP–MX–AG with a density of ~7 mg/cm$^2$ was lower than the one with a density of ~10 mg/cm$^2$ (Supplementary Fig. 28). This is because, at a lower mass density, fewer conductive pathways can be formed during compression, which results in a smaller resistance change under the same applied pressure[25].

To further evaluate the subtle pressure sensing durability and stability of BBP–MX–AG, the sensor was subjected to cyclic press–release experiments at pressures between 0 and 1 Pa. The output current of the sensor was maintained for over 10,000 cycling tests (Fig. 5a), and the specific isolated sensing curves at the 1st, 100th, 1000th, 2000th, 5000th, and 10,000th press–release cycles coincided with each other (Fig. 5b). Remarkably, the detection limit of 0.0063 Pa was retained, and a pressure sensitivity of more than 1800 kPa$^{-1}$ was maintained for pressures below 0.02 Pa, even after 10,000 press–release cycles (Fig. 5c). These results confirm the good monotonicity, as well as the long-term durability and stability of the BBP–MX–AG sensor. By contrast, the output current of the MX–AG sensor exhibited severe fluctuations during the 10,000 cycling tests in the 0–1 Pa range (Supplementary Fig. 29), indicating its poor sensing durability.

Finally, to demonstrate the practical uses of the subtle pressure-sensing of the BBP–MX–AG sensor, we applied the flexible piezoresistive sensor as a wearable sensor around a volunteer's neck to directly measure and monitor the central blood pressure waveforms and pulsations from the carotid artery (CA) and internal jugular vein (IJV) in a non-invasive manner (Fig. 6a). Central blood pressure waveforms can provide information regarding cardiovascular diseases or events[5,62]. The CA, which is located approximately 2.5 cm under the skin and carries abundant blood from the left ventricle and left atrium to the rest of the human body, can generate strong blood pressure waveform signals related to the left heart activity[4,5]. In contrast, the waveform signals of IJV pulses are extremely weak because the IJV is located deep inside the human neck[4,5]. In clinical practice, it is extremely difficult to continuously and accurately carry out the real-time monitoring of the IJV pulse waveform in a non-invasive manner[4,5]. Here, we demonstrate that our BBP–MX–AG sensor can provide an effective non-invasive approach for IJV monitoring.

Figure 6b presents the measurements of the CA and IJV using the BBP–MX–AG sensor. The specific measurement locations for the CA and IJV were approximately 9 and 6 cm away from the volunteer's collarbone, respectively. Two sequences of continuous waveforms were recorded for the CA pulses with strong pulsation signals (Fig. 6c) and the IJV pulse with weak pulsation signals (Fig. 6e). The high sensitivity of our BBP–MX–AG sensor in the subtle pressure range is illustrated in Fig. 6d, f, where a magnified view of a typical blood pressure waveform displays the detailed characteristic peaks and shapes of vessel pulses. The characteristic CA pulse waveform (Fig. 6d) exhibited three major components: the systolic phase, dicrotic notch, and diastolic phase, which can be interpreted to obtain valuable clinical information related to the left heart's activity[5]. Significantly, the unique shape of the IJV pressure waveform from a healthy subject can be obtained by our sensing device (Fig. 6f). The typical internal jugular vein pressure pulsation (Fig. 6f) features a biphasic waveform composed of three peaks: the A peak (right atrial contraction), C peak (right

ventricular contraction, leading to the bulging of the tricuspid valve to the right atrium), and V peak (venous filling and venous pressure increase), and two valleys: X (right ventricular systole, atrium relaxing) and Y (tricuspid valve opening and right ventricular filling)[4,5]. The stable, uniform, and identifiable waveform signals indicate that the BBP–MX–AG sensor can continuously and accurately monitor cardiovascular events in a non-invasive manner.

Moreover, to meet the requirements of future artificial intelligence, human–computer interfaces, and electronic skin applications, it is necessary to develop pressure-sensing arrays to realize spatial sensing distributions and tactile sensations. Therefore, we constructed a flexible 5 × 5 proof-of-concept pressure sensing array with an active pixel area of 5 × 5 mm$^2$ and a total device area of 3 × 3 cm$^2$ (Fig. 6g and Supplementary Fig. 30). The sensing array was fabricated by integrating 25 BBP–MX–AG monoliths (5 × 5 × 3 mm) on top of 25 interdigital silver electrodes on a PET substrate. An ultralight 3-cm-long hair with a weight of 0.21 mg was placed on the sensing array to carry out the pressure mapping test (Fig. 6g). The output current signals of the sensing pixels accurately distinguished minute weight differences and mapped the pressure distribution of the hair's position (Fig. 6h). Moreover, as demonstrated in Fig. 6i, the sensing array reliably and rapidly detected and tracked the movement of a mosquito (1.75 mg) landing on and taking off from the sensor's surface. We are not aware of any flexible pressure sensor with such high sensing ability that can detect such minimal dynamic and static loads. These results demonstrate the great potential for applying our sensing devices in ultrasensitive tactile sensing and electronic skin.

## Discussion

Here, we developed a piezoresistive aerogel by leveraging conductive MXene nanosheets and an ultrasoft bottlebrush polymer to assemble shrinkable nanochannel structures inside the multilevel cellular walls of an aerogel. This aerogel design presents considerable structural, electrical, and mechanical advantages for sensing subtle force signals. We show that the BBP–MX–AG aerogels are extremely sensitive piezoresistive sensors with remarkably low sensing limits, unprecedented high sensitivity for subtle pressure, rapid response time, high sensing resolution, and robust sensing durability. By demonstrating the feasibility of this design for cutting-edge applications, it is expected that our strategy will open new avenues for the development of piezoresistive materials with high sensitivity and low detection limits.

## Methods

**Raw materials.** Titanium aluminum carbide (Ti$_3$AlC$_2$) was purchased from 11 Technology Co., Ltd. (China). 3-Glycidyloxypropyldimethoxymethylsilane (GPDMS) and 3-glycidoxypropyltrimethoxysilane (GPTMS) were purchased from Meryer (Shanghai) Chemical Technology Co., Ltd. (Shanghai, China), with purity of 99.9%.

**Synthesis of Ti$_3$C$_2$T$_x$ nanosheets.** Titanium carbide (Ti$_3$C$_2$T$_x$) MXene was synthesized by the selective etching of aluminum in titanium aluminum carbide (Ti$_3$AlC$_2$, 400 mesh size) and delamination using minimally-intensive layer delamination (MILD)[63]. First, lithium fluoride (2 g) and hydrochloric acid (40 mL, 9 M) were mixed via stirring for 30 min. Then, Ti$_3$AlC$_2$ (1 g) was added to the solution. After heating at 35 °C with stirring for 24 h, the solution was poured into a centrifuge tube and then centrifuged at 3500 rpm for 10 min. The supernatant was removed and distilled water was added, and these two steps were repeated until the pH of the supernatant was >5. Then, the Ti$_3$C$_2$T$_x$ layers precipitated at the bottom were dispersed in ethanol by ultrasonication for 30 min. Then, ethanol was removed by centrifugation at 16,770 × g for another 30 min. Next, with the addition of distilled water, the remaining precipitate was centrifuged at 3500 rpm to form a uniform Ti$_3$C$_2$T$_x$ suspension. Finally, the resulting homogeneous MXene solution was freeze-dried to obtain MXene powder.

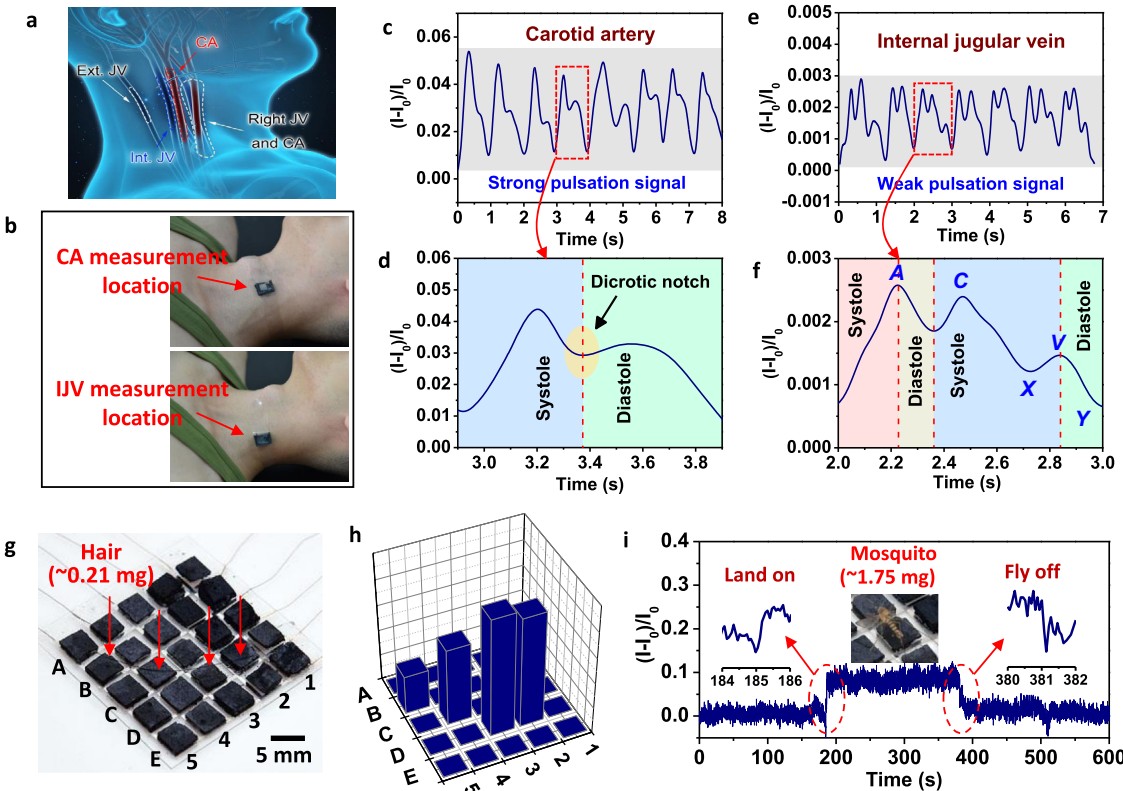

**Fig. 6 Application of piezoresistive BBP–MX–AG sensors for detecting subtle force signals. a** Illustration of the locations of the carotid artery (CA) and internal jugular vein (IJV) in a human's neck. **b** Photographs showing the sensing device mounted on specific locations of a human subject's neck to monitor CA and IJV pulses. The measurement locations for CA and IJV were approximately 9 and 6 cm away from the subject's collarbone, respectively. **c** Current response and **d** detailed interpreted waveforms of the CA pulse measured from a volunteer. **e** Current response and **f** detailed interpreted waveforms of the IJV pulse measured from a volunteer. The current signals measured from the IJV pulse were more than one order of magnitude smaller than those from the CA pulse. **g** Photograph of a 5 × 5 pixel sensing array detecting a human hair weighing ~0.21 mg (~3 cm in length), and (h) the corresponding distribution of the normalized current change on the sensing array. **i** Real-time current response to a mosquito (1.75 mg) landing on and taking off from the sensing array surface.

**Preparation of BBP–MX–AG.** First, an MXene solution (10 mg/mL) was prepared by dispersing $Ti_3C_2T_x$-MXene powder in distilled water via oscillation. Then, an appropriate amount of GPDMS was added to the solution with oscillation for 10 min. The mass ratio of MXene-to-GPDMS was 100:20. The solution was sealed in a hydrothermal reactor and heated for 1 h at 80 °C. Next, the obtained solution was poured into a designed container and placed in a cold source at a uniform temperature until frozen completely[32]. After freeze-drying for 3–5 days, ice crystals were volatilized completely, and the sample was removed from the container. Finally, the samples were delivered into a tube furnace and annealed at 200 °C in an argon atmosphere for 2 h. ICP–MX–AG was fabricated in the same way, but GPTMS replaced GPDMS. The density of the aerogel was calculated according to the equation $\rho = m/v$, where $\rho$ is the density, $m$ is the mass of the aerogel (obtained by weighing), and $v$ is the volume of the aerogel (calculated by measuring the length, width, and height of the aerogel sample).

**Preparation of MX–AG.** First, an MXene solution (10 mg/mL) was prepared by dispersing MXene powder in distilled water via oscillation. Next, the obtained solution was poured into a designed container and then placed in a cold source with a uniform temperature until it was completely frozen. The MXene aerogel was obtained after freeze-drying.

**Characterization and measurements.** Compression experiments were performed using a compression testing machine (AGS-X 5 N, Shimadzu, Japan) at a strain rate of 5 mm/min. Before the compression test, the sample was pre-compressed with 80% compression. The Young's modulus was calculated during the first 40% of the compression process. The energy loss coefficient was calculated by dividing the work loss by the compression work. The value of work was determined by the following formula: $W = \int_{\varepsilon_1}^{\varepsilon_2} \sigma d\varepsilon$, where $W$ is the specific work, $\varepsilon_1$ and $\varepsilon_2$ are the initial and final compressive strains, respectively, and $\sigma$ is the compressive stress[64].

To evaluate the pressure sensing performance, a pressure sensor was constructed by integrating a monolithic aerogel on top of a flexible interdigital electrode-coated PET substrate. Subtle pressure stimuli were applied in two ways: (1) using a mechanical testing stage (AGS-X 5 N, Shimadzu, Japan) equipped with a force sensor with an ultrasensitive force gauge (with a limit of 0.025 Pa); (2) using a loudspeaker box whose precise volume was determined by a decibel meter. A computer-controlled homemade loudspeaker box was used to produce sound or acoustic waves with various frequencies. The sound source was positioned 5 cm over the sensing device and was faced directly at the sensor surface. The output sound wave is edited by "GoldWave" software, and the sound wave template is in "freq" mode under "wave". The volume of the output acoustic signal was controlled by computer, and the precise volume or sound level reaching the sensor surface was calibrated and determined by a decibel meter. The resistance and current changes were measured using a Keithley 2000 digital multimeter. To prevent environmental noise interference, both the sound source and pressure sensor were installed inside a noise isolation chamber. The sound pressure ($P$) was defined as $P = P_0 10^{(\frac{L_{dB}}{20})}$, where $P_0$ and $L_{dB}$ denote the reference sound pressure in air (20 μPa) and the measured sound pressure level, respectively[65].

The sensing arrays were composed of 25 BBP-MX-AG monoliths with dimensions of $5 \times 5 \times 3$ mm³. The electrodes with five crossing horizontal lines and five vertical lines were vacuum-evaporated onto a PET substrate, and the intersection points were isolated from each other through the PDMS film. Then, the aerogels were attached to the points and connected to Keithley 2700 digital multimeter through a copper wire for further data collection.

The morphology of the aerogel was characterized by SEM (JSM-7800, Japan). HRTEM images were obtained using a transmission electron microscope (FEI Tecnai G2F30, 300 kV). XRD measurements were performed using a Rigaku Smart Lab 3 kW diffractometer (Rigaku, Japan). Fourier-transform infrared spectroscopy was performed using a Bruker TENSOR27 FT-IR spectrometer in the 4000–400 cm⁻¹ range. XPS characterization was performed using an ESCALAB 250Xi (Thermo Scientific, USA).

The in-situ TEM samples were prepared by focused ion beam using an ultrahigh-resolution dual-beam scanning electron microscopy system (TESCAN GAIA3 model 2016). During in situ TEM (JEM-ARM200CF, 200 kV) observations, an external force was applied using a tungsten blunt head operated by a NanoFactory system (EP1000). The distance change between the layers under pressure was analyzed by Digital Micrograph 1.6.2 software.

To avoid unexpected movements during IJV measurements, a volunteer was instructed to lie still on a chair or bed with controllable lying angles during the test. Then, to remove the interference from breathing signals and precisely extract the carotid pulse signals and IJV waveforms from the retrieved data, the as-measured waveforms were filtered by fast Fourier-transform with a bandpass filter from 1.0 to 4.0 Hz

## Data availability

The experiment data that support the findings of this study are available from the corresponding authors upon requests.

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

## Acknowledgements

This work reported here was supported by the National Natural Science Fund of China (51872146, 52173238, 51633002, 11872063, and 12172346), the Municipal Natural Science Fund of Tianjin (20JCJQJC00010), the National Key Research and Development Program of China (2016YFA0200200, and 2021YFA0715700), the Fundamental Research Funds for the Central Universities of Nankai University (63191520), and the Youth Innovation Promotion Association CAS.

## Author contributions

J.L. supervised the project. J.L. and Y.C. conceived and designed the research. X.S., X.F., Y.L., P.W., X.J. participated in materials preparation, device fabrication, device test or interpretation of results. Y.Z., B.W., H.-A.W, performed the mechanical simulations and analyzations, R.J., H.Z., J.W. contributed the in situ HRTEM characterizations, J.L., Y.C., X.S. analyzed the data and co-wrote the manuscript. All authors analyzed and discussed the results.

## Competing interests

The authors declare no competing interests.
