## [Peer Review File · Nature Communications]

REVIEWER COMMENTS

Reviewer #1 (Remarks to the Author):

In this manuscript, the authors have proposed a rational “3C” design strategy to fabricate a bottlebrush polysiloxane cross-linked MXene aerogel which possesses a multilevel nano-channels inside its cellular walls for ultrasensitive subtle pressure sensing. This well-designed piezoresistive aerogel material, featuring low Young’s modulus of 140 Pa, numerous variable conductive pathways, and mechanical robustness, can detect subtle pressure signals of 0.0063 Pa, deliver brilliant pressure sensitivity larger than 1900 kPa^{-1} under subtle pressure range below 0.02 Pa, and exhibit impressive sensing robustness. These findings are vital to promote the development of ultrasensitive tactile sensors and electronic skins. The paper was well-prepared, and the experimental results are clear and convincing. Some issues are required to address to further improve the manuscript before publication.

1. The size for one pixel ($5 \times 5 \times 3 \text{ mm}$) in the sensing array as shown in Figure 6e is too large to construct a high sensing density. A sensing array with smaller pixels and higher sensing resolution should be provided.
2. Why the sensing curve of BBP-MX-AG in figure 4f show a two-stage increase with the increase of applied pressure?
3. Why the maximum compressive strain in Figure S10 for the MX-AG was only 30%?
4. Whether the BBP-MX-AG with a density of $\sim 7 \text{ mg/cm}^2$ and Young’s modulus of 58 Pa (Figure 2e) can exhibit an even smaller pressure detectable limit?
5. In figure 6d, while the IJV waveform signals were ultra-weak, how to avoid the interference signals from unexpected movement or activity?
6. Except for the foam-like structure, the strain sensors could also exist in the form of densely packed film, for example, the publication in Science Bulletin, 2020, 65(11): 899, what is the difference between these pressure sensors with different structures?

Reviewer #2 (Remarks to the Author):

This paper claims “3C” design strategy for fabrication polysiloxane cross-linked MXene aerogel with ultralow Young’s modulus and excellent mechanical robustness. The aerogel can detect subtle pressure signals of 0.0063 Pa. This manuscript just test some properties of MXene aerogel, but the mechanism and real structure of MXene aerogel are not revealed. Too many assumptions in the manuscript but no any direct evidence for revealing the structure of MXene aerogel. Meanwhile, there are many low-class errors in the manuscript. So this work cannot reach quality of Nature Communications. Many issues should be carefully addressed:

1. What are basic design requirements? Are there any reports or references?
2. There are many low-class mistakes, such as the sentence in lines 92-94 "Previous work has reported that in cellular walls^{13, 14}." should be corrected into "Previous works have reported that in cellular walls^{13, 14}." Please carefully double check the main text.
3. Please cited the reported papers in your sentences such as "Overall, although the recent intense research effort has effectively advanced the development of piezoresistive aerogels and achieved high performance for certain aspects of sensing properties." In lines 102-104. Do not forejudge anything in manuscript without evidence or references!
4. What is the intrinsic sense of "3C" or just the abbreviation of words (Composite materials, nano-channel structure, cross-linking interface)? Besides, there are any logical relationships between these three words. The authors obviously oversell the term "3C".
5. What is the words for abbreviation of BBP-MX-AG in line 113? What does BBP represents? And MX, AG? This abbreviation looks too arbitrary!
6. The cartoon images in Fig 1 looks beautiful, however, there is no any direct evidence such as SEM, TEM, Micro-CT images for verifying the structure of cartoon images. Obviously, these assumptions are not reasonable!
7. Many references are wrongly cited, such as "...covalently bond with MXene nanosheets via the formation of Ti-O-Si¹²¹" in line 172 (The MXene is first reported in 2011, however, the ref. 21 has already published in 2001.); "PGPDMS covalently cross-links the MXene nanosheets va the formation of Ti-O-Si bonds (Supporting Note 3)²²." (There is no any words about the Ti-O-Si covalent bond between MXene and SiO₂ in ref. 22. It is completely wrong citation.)
8. The authors claim that "PGPDMS can act as a spacer materials for intercalation into the MXene interlayer and prevent the restacking of the MXene nanosheets via molecular assembly." So how to verify this conclusion?
9. The authors have make numerus assumptions such as "Such multilevel nano-channel structure allows the detection of tiny external force by shrinking (or expanding) the space between the nano-channels (meeting requirement (i)). Moreover, numerous new conductive pathways can be created between the neighboring MXene nanosheets during nano-channel shrinking (meeting requirement (ii)), leading to considerable resistance changes inside the cellular walls....." in lines 178-183.
10. The authors have claimed that "ICP-MX-AG are investigated using molecular dynamics (MD) simulations (Figure S4) and Supporting note 2)". However, there is no any direct evidence for verifying the structure of ICP-MX-AG, so what is the sense of simulation? What is the base for using MD simulation?
11. The density is very important parameter! So the calculation procedure details should be provided.
12. The authors claim that ".....expansion of nano-channels in the cellular walls of BBP-MX-AG" in line 256, however, there is no any direct evidence to verify this conclusion. The in-situ characterization only just shows the curving of MXene bundle sheets under loading.

Reviewer #3 (Remarks to the Author):

Here the composite aerogel by Shi et al. could respond to subtle stimuli down to 0.0063 Pa with a high sensitivity over 1400/kPa. The result, of course, is impressive. However, I do have concerns on the novelty of this work. Please address the following points for further submission.

1. The authors stressed high sensitivity of their sensor. However, high sensitivity at low pressures is not a challenging work. There have been many published papers reporting a maximum sensitivity far higher than 1900/kPa. For example, ACS Nano 2021, 15, 1795–1804 (sensitivity= 380000/kPa); and ACS Appl. Mater. Interfaces 2018, 10, 40880–40889. The authors should do a more comprehensive research on the state of the art of this field.
2. The demonstration of the change in the d-spacings of the MXene layers via in-situ TEM inspection seems amazing (Figure 3). However, I question about the applied stress level in this observation (by using indentation). Since the specimen is significantly deformed, the stress is estimated to be on GPa level, while the sensing test is conducted on 0.001-1 Pa level. The difference is at about 10 orders of magnitude. Obviously, it is not convincing to clarify the sensing mechanism by using the in-situ indentation and TEM observation.
3. The authors claimed that: To realize accurate, continuous, and ultrasensitive monitoring of ultraweak pressure stimuli, a desired piezoresistive material should have all of the following structural and material properties: (i) ultralow Young's modulus to significantly reduce the critical stress value that triggers the material deformation; (ii) multilevel or hierarchical structure with ultrahigh densities of variable conductive pathways to allow very large changes in the electrical conductivity of the sensing materials during structural deformation; (iii) excellent mechanical elasticity and robustness to prevent the collapse and disintegration of the material during repeated structure deformations. I agree that a low Young's modulus may help. However, there should be more ways to achieve ultrasensitivity other than what are claimed here. Also, the second and the third points are not convincing.
4. Applying a small pressure lower than 0.1 Pa is difficult. I like the method offered by the authors. They may provide more details on the use of acoustic pressure.
5. There are too much typo and grammatic errors. They should ask a native English speaker to further improve the language.

REPLY to REVIEWERS
for MANUSCRIPT NCOMMS-21-05897-T

Dear Editor:

We appreciate your consideration of our manuscript! We also appreciate the valuable comments and advises from the reviewers, which can help to improve the quality of our manuscript! We are hereby submitting the revised manuscript entitled “*Pushing the Detectability and Sensitivity for Subtle Force Signals to New Limits with a Shrinkable Nanochannel Structured Composite Aerogel*” (NCOMMS-21-05897-T) for your consideration for publication in *Nature Communications*. Below are our responses to the reviewers’ comments. We have made major revisions to our manuscript according to the comments, and all revisions are highlighted in the revised manuscript.

We do hope the paper after revision would find your approval for publication. Your kind consideration is greatly appreciated. We are looking forward to hearing from you soon.

Sincerely,

Jiajie Liang & Yongsheng Chen

Professor of Materials Science and Engineering

A list of changes highlighted in the revised manuscript and supplementary information:

1. A sensing array with smaller pixels and higher sensing resolution had been provided in Supplementary Figure S30 according to the comments from reviewer#1.
2. The sensing performance of BBP-MX-AG with a density of 7 mg/cm² had been added (on Page 18 and Supplementary Figure S28) according to the comments from reviewer#1.
3. The detailed testing conditions had been added in Method part (on Page 27 and 28) according to the comments from reviewer#1.
4. The “*basic design requirements*” and “3C” had been deleted, and the introduction part, abstract, and conclusion had been rewritten in the revised manuscript (on Page 3, 4, and 5) according to the comments from reviewer#2 and reviewer#3.

5. More references had been added to support our statements and conclusion (on Page 3, 4, 5, 8, 11, 14, 17, 21, and 22) according to the comments from reviewer#2.
6. The abbreviations had been improved according to the comments from reviewer#2 (on Page 7, 8, and 9).
7. GPC analysis (Supplementary Figure S3), EDS mapping (Supplementary Figure S11), XRD patterns (Supplementary Figure S19), and more *in-situ* HRTEM images (Supplementary Figure S18) had been added to confirm the shrinkable multilevel nanochannel structure in the cellular walls of BBP-MX-AG according to the comments from reviewer#2.
8. More references had been cited to confirm the formation of Si-O-Ti bonds in BBP-MX-AG according to the comments from reviewer#2.
9. GPC analysis (Supplementary Figure S6), XRD patterns (Supplementary Figure S10), and FT-IR characterizations (Supplementary Figure S13) had been added to confirm the structure of ICP-MX-AG according to the comments from reviewer#2.
10. The density calculation details had been added in the Method part (on Page 25) according to the comments from reviewer#2.
11. Figure 3 had been modified and more *in-situ* HRTEM images (Supplementary Figure S18) had been added to confirm the shrinkable nature of nanochannel structure in the cellular walls of BBP-MX-AG according to the comments from reviewer#2.
12. More reference had been added in Supplementary Table 1 according to the comments from reviewer#3.
13. XRD patterns (Supplementary Figure S19) had been added to further confirm the shrinkable nature of nanochannel structure in the cellular walls of BBP-MX-AG under subtle pressure according to the comments from reviewer#3.
14. More details on the use of acoustic pressure had been added in the Method part (on Page 26) according to the comments from reviewer#3.
15. The language has been fully polished by a native English speaker according to the comments from reviewer#3.

Reply to the reviewer#1:

Comments: In this manuscript, the authors have proposed a rational “3C” design strategy to fabricate a bottlebrush polysiloxane cross-linked MXene aerogel which possesses a multilevel nano-channels inside its cellular walls for ultrasensitive subtle pressure sensing. This well-designed piezoresistive aerogel material, featuring low Young’s modulus of 140 Pa, numerous variable conductive pathways, and mechanical robustness, can detect subtle pressure signals of 0.0063 Pa, deliver brilliant pressure sensitivity larger than 1900 kPa⁻¹ under subtle pressure range below 0.02 Pa, and exhibit impressive sensing robustness. These findings are vital to promote the development of ultrasensitive tactile sensors and electronic skins. The paper was well-prepared, and the experimental results are clear and convincing.

REPLIES: We appreciate the positive comments!

Comments: The size for one pixel ((5 × 5 × 3 mm) in the sensing array as shown in Figure 6e is too large to construct a high sensing density. A sensing array with smaller pixels and higher sensing resolution should be provided.

REPLIES: Thanks for the suggestion! We have made a 5 × 5 sensing array with an active pixel area of 2 mm in diameter and the total device area of 1.5 × 1.5 cm² as demonstrated in Figure S30 in the revised Supplementary information (Highlight on Page 23 in the manuscript and Supplementary Figure S30).

Comments: Why the sensing curve of BBP-MX-AG in figure 4f show a two-stage increase with the increase of applied pressure?

REPLIES: Thanks for the question! This two-stage increase with the increase of applied pressure in the sensing curve in Figure 4f is attributed to two different sensing mechanism in the BBP-MX-AG. According to the equation S1 (Supplementary Note 1 in Supplementary

information), the critical stress value (σ_c) that triggers the bending or buckling of cellular walls in the BBP-MX-AG is calculated to be ~ 0.11 Pa. This indicates that the resistance changes in the pressure range below 0.1 Pa are mainly caused by the shrinking of the nano-channel structure inside the cellular walls (R_i) of BBP-MX-AG. When the applied pressure is greater than 0.1 Pa, the bending or buckling of cellular walls starts to contribute the resistance changes (R_E) in the BBP-MX-AG. Thus, a two-stage increase in the slope can be observed for the sensing curve as the applied pressure increases (Figure 4f).

Comments: Why the maximum compressive strain in Figure S10 for the MX-AG was only 30%?

REPLIES: Thanks for the question! Pure MXene aerogel (MX-AG) without the addition of polysiloxane exhibited poor mechanical properties, and the aerogel would collapse when the compressive strain was larger than 30%.

Comments: Whether the BBP-MX-AG with a density of ~ 7 mg/cm² and Young's modulus of 58 Pa (Figure 2e) can exhibit an even smaller pressure detectable limit?

REPLIES: Thanks for the question! Piezoresistive BBP-MX-AG (density = ~ 7 mg/cm²) exhibited nearly the same pressure detection limit (0.0063 Pa) as the one with a density of ~ 10 mg/cm² (Supplementary Figure S28). This is because the pressure detection limit is mainly determined by the shrinkable nanochannels in the cellular walls of BBP-MX-AG. Both BBP-MX-AG samples have similar multilevel nanochannels in their cellular walls because their initial MXene-to-silane weight ratio and fabrication conditions were the same; however, the sensitivity of BBP-MX-AG with a density of ~ 7 mg/cm² was lower than the one with a density of ~ 10 mg/cm² (Supplementary Figure S28). This is because, at a lower mass density, fewer conductive pathways can be formed during compression, which results in a smaller resistance change under the same applied pressure. We have added the corresponding data and discussion in the revised manuscript and Supplementary information (Highlight on Page 18 in the manuscript and Supplementary Figure S28).

Comments: In figure 6d, while the IJV waveform signals were ultra-weak, how to avoid the interference signals from unexpected movement or activity?

REPLIES: Thanks for the question! First, to avoid the unexpected movement, the volunteer was lying still on a chair or bed with controllable lying angles during the test. Then, to remove the interference from breathing signal and precisely extract the carotid pulse signals and IJV waveforms from the retrieved data, the as-measured waveforms were filtered by fast Fourier transform with a band pass filter from 1.0 to 4.0 Hz. We have added these experimental details in the Method part of the revised manuscript (Highlight on Page 27 and 28 in the manuscript).

Comments: Except for the foam-like structure, the strain sensors could also exist in the form of densely packed film, for example, the publication in Science Bulletin, 2020, 65(11): 899, what is the difference between these pressure sensors with different structures?

REPLIES: Thanks for the question! In fact, strain sensor and pressure sensor are two different types of electromechanical sensors. Pressure sensors transduce applied pressure into an electrical signal, facilitating the detection of the magnitude and/or direction of the pressure by measuring the pressure signal changes. For instance, pressure sensor can be applied to detect the pressure produced by human-body activity distributed from a low-pressure regime (<1 kPa) to a medium-pressure regime (<10 kPa) to a high-pressure regime (>10 kPa) [*Adv. Mater.* **2020**, 2003014]. The sensing mechanism of pressure sensors includes piezoelectricity, capacitance, and piezoresistivity, and the sensitivity of a pressure sensor is defined as the slope of relative electrical signal change versus the applied pressure.

By contrast, strain sensors (such as the sensing device reported in *Science Bulletin*, **2020**, 65(11): 899) transduce mechanical deformation into electrical signal when stretching the sensing devices. The sensitivity (gauge factor) of strain sensors refers to the slope of the curve of relative electrical signal change versus applied strain. Cracks generating and propagating in conductive thin film/network during stretching and thus greatly limiting the electrical conduction through

the thin film/network is the main mechanism exploited in strain sensors. The strain sensors, which was in the form of densely packed film reported in *Science Bulletin*, **2020**, 65(11): 899, is based on this crack-propagation mechanism.

Reply to the reviewer#2:

Comments: What are basic design requirements? Are there any reports or references?

REPLIES: Thanks for the question! We have deleted the corresponding statements of “*basic design requirements*”, reorganized the ideas, and rewritten the introduction part in the revised manuscript. In addition, more comprehensive references were cited to support our statement and analysis (Highlight in abstract and summary part, and on Page 3, 4, and 5 of the manuscript).

Comments: There are many low-class mistakes, such as the sentence in lines 92-94 “Previous work has reported that in cellular walls13, 14.” should be corrected into “Previous works have reported that in cellular walls13, 14.” Please carefully double check the main text.

REPLIES: Many thanks for pointing out the problems! In fact, this sentence indeed only introduced one work (*Adv. Mater.* **28**, 194-200 (2016)). We have deleted the other reference cited here. Also, we have double-checked the main text to avoid this mistake in the revised manuscript. Moreover, a native English speaker had helped us to fully polished the language of the revised manuscript.

Comments: Please cited the reported papers in your sentences such as “Overall, although the recent intense research effort has effectively advanced the development of piezoresistive aerogels and achieved high performance for certain aspects of sensing properties.” In lines 102-104. Do not forejudge anything in manuscript without evidence or references!

REPLIES: Thanks for pointing out this problem! We cited the reported papers in the corresponding sentences to support our statements in the revised manuscript (Highlight on Page 5 in the manuscript). Also, we have double checked the main text and more references were cited in the revised manuscript to support our statements and conclusion (Highlight on Page 3, 4, 5, 8, 11, 14, 17, 21, and 22 in the manuscript).

Comments: What is the intrinsic sense of “3C” or just the abbreviation of words (Composite materials, nano-channel structure, cross-linking interface)? Besides, there are any logical relationships between these three words. The authors obviously oversell the term “3C”.

REPLIES: Thanks for pointing out this problem! We have deleted this “3C” concept and revised the corresponding discussion and figure in the revised manuscript (Highlight on Page 3, 4, 5, and 6, and in the abstract and conclusion part of the manuscript).

Comments: What is the words for abbreviation of BBP-MX-AG in line 113? What does BBP represents? And MX, AG? This abbreviation looks too arbitrary!

REPLIES: Many thanks for the question, but we do not agree with the reviewer’s comment that “This abbreviation looks too arbitrary”. However, we still modified the corresponding sentences in the revised manuscript to make these abbreviations much clearer (Highlighted on Page 7, 8, and 9 in the manuscript). The abbreviations used in the manuscript are list below:

BBP is the abbreviation of **B**ottle**B**rush **P**GP**D**MS;

ICP is the abbreviation of **I**nterchain-**C**rosslinked **P**GP**T**MS;

MX represents **M**Xene;

AG is the abbreviation of **A**ero**G**el;

MX-AG represents pure **M**Xene **A**ero**G**el;

BBP-MX-AG is the abbreviation of **B**ottle**B**rush **P**GP**D**MS intercalated **M**Xene aerogel;

ICP-MX-AG is the abbreviation of **I**nterchain-**C**rosslinked **P**GP**T**MS intercalated **M**Xene aerogel.

Comments: The cartoon images in Fig 1 looks beautiful, however, there is no any direct evidence such as SEM, TEM, Micro-CT images for verifying the structure of cartoon images. Obviously, these assumptions are not reasonable!

Comments: The authors claim that “PGPDMS can act as a spacer materials for intercalation into the MXene interlayer and prevent the restacking of the MXene nanosheets via molecular assembly.” So how to verify this conclusion?

REPLIES: Thank for the questions! At first, we are a bit confused about the reviewer’s first comment “*The cartoon images in Fig 1 looks beautiful, however, there is no any direct evidence such as SEM, TEM, Micro-CT images for verifying the structure of cartoon images*”. But after we read the second comment “*The authors claim that “PGPDMS can act as a spacer materials for intercalation into the MXene interlayer and prevent the restacking of the MXene nanosheets via molecular assembly.” So how to verify this conclusion?*”, we think that the reviewer may question the shrinkable nanochannel structure of multilevel MXene layers intercalated and cross-linked with BBP. So, we tried to reply these two comments together here.

First, this polymer or organic molecular intercalated MXene-based (or graphene) multilevel nano-channel structure had been extensively investigated [Adv. Mater. 2018, 1804600; J. Mater. Chem. A, 2018, 6, 16196; Chem. Mater. 2020, 32, 1703; PNAS, 2014, 111, 16676; ACS Nano, 2019, 13, 649–659]. For instance, Zhao T. et al. reported the fabrication of hydrophobic Ti₃C₂ MXene membrane with enlarged spacing between the parallel MXene layers via intercalating and modifying MXene nanosheets with long chain of trimethoxy(1H,1H,2H,2H-per-fluorodecyl)silane [J. Mater. Chem. A, 2018, 6, 16196]. The authors utilized XRD and SEM measurements to characterize and prove their findings. Before silane modification, the pure MXene film exhibit tightly stacked multilayer structure; after silane modification, the composite film exhibited a loose layered structure with enlarged spacing between the parallel MXene layers, as confirmed by the SEM and XRD characterizations. **Moreover**, some recently published works had reported that Si-O-Ti linkage can be formed between silane and MXene via silylation reaction, and the existence of the Si-O-Ti bonding in the silane modified MXene composite was characterized by FT-IR and XPS [J. Mater. Chem. A, 2018, 6, 16196; Biosens. Bioelectron. 2018, 121, 243; Mater. Horiz. 2019, 6, 1057]. **In addition**, the shrinkable features of such multilevel nano-channel structure based on MXene and graphene had been characterized and observed by laser scanning confocal microscopy and in-situ HRTEM measurements [Adv. Mater. 2018, 31, 1804600; Nat. Commun. 2017, 8, 1207].

Following these typical characterizations approaches, in fact, we have provided reasonable

and direct evidence to verify the shrinkable multilevel nanochannel in the cellular walls of BBP-MX-AG in our original manuscript. First, as can be seen in the TEM characterizations in Figure 2b and 2c, the cellular walls in pure MX-AG without intercalation of polysiloxane exhibited compacted multilayered structure with interlayer distance of about 1.1 nm (Figure 2c), which is consistent with the typical interlayer distance for pristine $\text{Ti}_3\text{C}_2\text{T}_x$ films [J. Mater. Chem. A, 2018, 6, 16196; Chem. Mater. 2020, 32, 1703; PNAS, 2014, 111, 16676; ACS Nano, 2019, 13, 649–659]. After intercalating polysiloxane, the cellular walls of BBP-MX-AG also exhibited multilevel nano-channel structure with enlarged interlayer distance of ~1.8 nm (Figure 2b and Figure 3). Moreover, XRD patterns in Figure 2d also confirmed that the interlayer distance of the cellular walls in MX-AG and BBP-MX-AG is about 1.1 nm and 1.8 nm, respectively. These confirm the successful intercalation of polysiloxane into the MXene interlayer at the molecular level to form parallel nano-channel structure inside the cellular walls of BBP-MX-AG. Then, we further used FTIR and XPS characterizations to verify the covalently cross-linking between the intercalated PGPDMs and MXene nanosheets, as shown in Supplementary Figure S12, S13 and S4, and Supplementary Note 3 in the Supplementary information: “The covalent crosslinking between MXene and PGPDMs (or PGPTMS) was confirmed by Fourier-transform infrared (FTIR) spectra (Supplementary Figure S12 and S13) and X-ray photoelectron spectroscopy (XPS) (Supplementary Figure S14). Compared with pure MX-AG, three new peaks at 1245, 1197, and 465 cm^{-1} ascribed to Si-CH₂ and Si-O-Si stretching, and O-Si-O deformation, respectively⁷, appeared in the spectra of BBP-MX-AG (before annealing) and BBP-MX-AG. Compared with the spectrum of BBP-MX-AG (before annealing), a new characteristic Ti-O-Si peak at 942 cm^{-1} appeared in the spectra of BBP-MX-AG and ICP-MX-AG, indicating that the MXene nanosheets were covalently linked with PGPDMs and PGPTMS through Ti-O-Si hetero-linkages in BBP-MX-AG and ICP-MX-AG” Last, the shrinkable nature of the nano-channel structure inside the cellular walls of BBP-MX-AG was characterized by in-situ HRTEM as demonstrated in Figure 3 in the original manuscript. the thickness (t) of a cellular wall containing 5 MXene layers (indicated by the red line) decreased from 7.1 nm (1 s) to 5.01 nm (19 s) when an external force was applied, and then recovered to 6.87 nm (28 s) when the stimulating force was released. This cellular wall thickness change was induced by a simultaneous spacing change in the nanochannels of the cellular walls, as confirmed by digital micrograph images (Figures 3e-h). Specifically, the average distance (d_A)

between two MXene layers decreased from an initial value of 1.78 nm to 1.6 nm and 1.25 nm at 15 s and 19 s sequentially during the loading process, which then increased back to 1.72 nm at 28 s when unloaded (Supplementary Movie 1). **Thus, we have provided decent evidence to verify the shrinkable nano-channel structures as shown in the cartoon images in Figure 1.**

Moreover, to further verify our conclusion, more characterizations were carried out and added in the revised manuscript and Supplementary information. We added Gel Permeation Chromatography (GPC) analysis to verify that polysiloxane (PGPDMS) with M_w about 7350 can be obtained from the fabrication process of BBP-MX-AG (Highlight in Supplementary Figure S3). Elemental (EDS) mapping was carried out to reveal the presence and detailed spatial distribution of elemental Si (from PGPDMS) over a large area BBP-MX-AG (Highlight in Supplementary Figure S11). To further verify the shrinkable nano-channel structure in the cellular walls of BBP-MX-AG, we characterize the change of interlayer distance of the cellular walls in BBP-MX-AG under tiny force using XRD patterns, and the corresponding discussion and data were added in the revised manuscript and SI (Highlight on Page 13 and 14 in the manuscript and Supplementary Figure S19). As shown in XRD patterns of Supplementary Figure S19 in the revised SI, the interlayer distance of the nano-channel structure in cellular walls of BBP-MX-AG decreased from the original value of 1.8 nm (without any external force) to the values of about 1.5 nm and 1.4 nm when external pressure of 0.2 Pa and 1 Pa were supplied on the sample, respectively, indicating the decrease of interlayer distance as the increase of applied pressure.

Thus, all these characterization results verify the hierarchical nanostructure of PGPDMS intercalated multilevel cellular walls in the BBP-MX-AG as illustrated in Figure 1a.

Comments: Many references are wrongly cited, such as "...covalently bond with MXene nanosheets via the formation of Ti-O-Si121" in line 172 (The MXene is first reported in 2011, however, the ref. 21 has already published in 2001.); "PGPDMS covalently cross-links the MXene nanosheets va the formation of Ti-O-Si bonds (Supporting Note 3)22." (There is no any words about the Ti-O-Si covalent bond between MXene and SiO2 in ref. 22. It is completely wrong citation.)

REPLIES: Thanks for pointing out this problem! We have cited the corresponding works in the revised manuscript [*J. Mater. Chem. A*, 2018, 6, 16196; *Biosens. Bioelectron.* 2018, 121, 243; *Mater. Horiz.* 2019, 6, 1057]. However, we want to point out that the citations of ref 21 and 22 cited in our original manuscript were not “*completely wrong citation*”. When we started the research about MXene functionalization in 2017, few papers had reported that MXene could be silanized with silane agent. After carefully checking a number of papers (including ref 21 and 22 cited in our original manuscript), we thought that the surface functional groups of Ti-OH on MXene nanosheets could be reacted with Si-OH to form Si-O-Ti bonds via silylation reactions. This is because Ti-OH groups on TiO₂ particles can form Ti-O-Si linkages with Si-OH (derived from the hydrolysis of silane) via silylation reaction (ref 21 and 22 cited in our original manuscript). Moreover, the successful formation of Si-O-Ti bonds between MXene and PGPDMS have been directly characterized in our original manuscript through FT-IR (Supplementary Figure S12) and XPS (Supplementary Figure S14) characterizations in the Supplementary information.

Comments: The authors have make numerus assumptions such as “Such multilevel nano-channel structure allows the detection of tiny external force by shrinking (or expanding) the space between the nano-channels (meeting requirement (i)). Moreover, numerous new conductive pathways can be created between the neighboring MXene nanosheets during nano-channel shrinking (meeting requirement (ii)), leading to considerable resistance changes inside the cellular walls.....” in lines 178-183.

REPLIES: Thanks for the comments! We have deleted these sentences, rewritten corresponding statements, revised the discussion (Highlight in abstract and summary part, and on Page 3, 4, and 5 of the manuscript), and cited more reported papers to support our conclusion in the revised manuscript [*Adv. Mater.* 2018, 1804600; *Nat. Commun.* 2017, 8, 1207; *Adv. Funct. Mater.* 2020, 1909603]. As discussed above, we have verified the presence of shrinkable nano-channel structure of multilevel MXene layers intercalated and cross-linked with BBP in the BBP-MX-AG. When pressure was applied to BBP-MX-AG, the interlayer space between the neighboring MXene nanosheets in the cellular walls would decrease simultaneously, leading to the reduce of

internal resistance R_I and increase of the conductivity for the BBP-MX-AG. A tiny external pressure thus can be monitored via the resistivity changes originated from the changed interlayer distance [Adv. Mater. 2018, 1804600; Nat. Commun. 2017, 8, 1207; Adv. Funct. Mater. 2020, 1909603]. By contrast, when pressure was applied to MX-AG, the interlayer space between the neighboring MXene nanosheets would not change due to the compacted multilayered structure in the cellular walls, and the internal resistance R_I did not change. Since the sensitivity of a pressure sensor is defined as the slope of relative resistance change versus the applied pressure, BBP-MX-AG can exhibit larger R_I change and higher sensitivity than MX-AG under pressure.

Comments: The authors have claimed that “ICP-MX-AG are investigated using molecular dynamics (MD) simulations (Figure S4) and Supporting note 2)”. However, there is no any direct evidence for verifying the structure of ICP-MX-AG, so what is the sense of simulation? What is the base for using MD simulation?

REPLIES: Thanks for pointing out this problem! First, we have added the scheme of synthetic route of ICP-MX-AG and BBP-MX-AG in the revised supplementary information (Highlight in Supplementary Figure S2 and S5). Then, GPC analysis was carried out and added in the revised SI to verify that polysiloxane (interchain-crosslinked PGPTMS) with M_w about 8724 can be obtained from the fabrication process of ICP-MX-AG (Highlight in Supplementary Figure S6). Next, XRD characterization was carried out and added in the revised SI to confirm the success of intercalation of Interchain-Crosslinked PGPTMS into the interlayer of MXene based nano-channel structure (Highlight in Supplementary Figure S10). The interlayer distance of the cellular walls in MX-AG and ICP-MX-AG is about 1.1 nm and 1.8 nm, respectively. Last, we further carried out the FT-IR characterizations to verify the covalently cross-linking between the intercalated Interchain-Crosslinked PGPTMS and MXene nanosheets (via Ti-O-Si bonding), as highlighted in the Supplementary Figure S13 in the revised Supplementary information. From all these characterizations, we can confirm the structure of ICP-MX-AG.

Comments: The density is very important parameter! So the calculation procedure details should

be provided.

REPLIES: Thanks for the question. The density of the aerogel was calculated according to the equation $\rho = m/v$, where ρ is the density, m is the mass of the aerogel (obtained by weighing), and v is the volume of the aerogel (calculated by measuring the length, width, and height of the aerogel sample). We have added this detail in the experimental part of the revised manuscript (Highlight on Page 25 in the manuscript).

Comments: The authors claim that “.....expansion of nano-channels in the cellular walls of BBP-MX-AG” in line 256, however, there is no any direct evidence to verify this conclusion. The in-situ characterization only just shows the curving of MXene bundle sheets under loading.

REPLIES: Thanks for the question. Please carefully check the thickness changes in the cellular walls **labeled by red lines** in the in-situ HRTEM images (Figure 3a-d in the original manuscript). The thickness (t) of a cellular wall containing 5 MXene layers (indicated by the red line) decreased from 7.1 nm (1 s) to 5.01 nm (19 s) when an external force was applied, and then recovered to 6.87 nm (28 s) when the stimulating force was released. This cellular wall thickness change was induced by a simultaneous spacing change in the nanochannels of the cellular walls, as confirmed by digital micrograph images (Figures 3e-h). Specifically, the average distance (d_A) between two MXene layers decreased from an initial value of 1.78 nm to 1.6 nm and 1.25 nm at 15 s and 19 s sequentially during the loading process, which then increased back to 1.72 nm at 28 s when unloaded (Supporting movie 1). Thus, this is the **direct** evidence to confirm the shrinking and expanding features of the nanochannel structure, which is consistent with the result observed in previously published work [*Nat. Commun.* 2017, 8, 1207]. We have modified Figure 3 in the revised manuscript to make it much clearer. Moreover, we also added more magnified HRTEM images in the revised Supplementary information to confirm this shrinkable nanostructure (Highlight in Supplementary Figure S18).

Reply to the reviewer#3:

Comments: Here the composite aerogel by Shi et al. could respond to subtle stimuli down to 0.0063 Pa with a high sensitivity over 1400/kPa. The result, of course, is impressive.

REPLIES: Thanks for the positive comments!

Comments: The authors stressed high sensitivity of their sensor. However, high sensitivity at low pressures is not a challenging work. There have been many published papers reporting a maximum sensitivity far higher than 1900/kPa. For example, ACS Nano 2021, 15, 1795–1804 (sensitivity= 380000/kPa); and ACS Appl. Mater. Interfaces 2018, 10, 40880–40889. The authors should do a more comprehensive research on the state of the art of this field.

REPLIES: Thanks for pointing out this problem! We have done a more comprehensive research on the state of the art of this field and added the corresponding works in Supplementary Table 1 in the revised Supplementary information for comparison (**Highlight in Supplementary Table 1**). Although the sensitivity of our BBP-MX-AG sensor was lower than that of the devices reported in ACS Nano 2021, 15, 1795 and ACS Appl. Mater. Interfaces 2018, 10, 40880, the detectable pressure limit of our sensing device (0.0063 Pa) was still better than that of the sensing devices published in ACS Nano 2021, 15, 1795 (0.025 Pa) and ACS Appl. Mater. Interfaces 2018, 10, 40880 (5 Pa).

Comments: The demonstration of the change in the d-spacings of the MXene layers via in-situ TEM inspection seems amazing (Figure 3). However, I question about the applied stress level in this observation (by using indentation). Since the specimen is significantly deformed, the stress is estimated to be on GPa level, while the sensing test is conducted on 0.001-1 Pa level. The difference is at about 10 orders of magnitude. Obviously, it is not convincing to clarify the sensing mechanism by using the in-situ indentation and TEM observation.

REPLIES: Thanks for the question! Although the specific value of the external force supplied by the nano-indenter during the in-situ HRTEM characterization in Figure 3 was unclear, it should not be at the GPa level since the critical stress value that triggers the buckling or bending of the cellular walls in the aerogels was lower than 1 Pa, as calculated in the Supporting note 1 in the original SI. To verify the shrinkable nanochannel structure in the cellular walls of BBP-MX-AG, we further characterize the change of interlayer distance of the cellular walls in BBP-MX-AG under tiny force using XRD, and the corresponding discussion and data were added in the revised manuscript and SI (Highlight on Page 13 and 14 in the manuscript and Supplementary Figure S19). As can be observed from the XRD patterns in Figure S19, the interlayer spacing of the nanochannels in the cellular walls of BBP-MX-AG decreased from an original value of 1.8 nm (without an external force) to 1.5 nm and 1.4 nm when external pressures of 0.2 Pa and 1 Pa were applied to the sample, respectively.

Comments: The authors claimed that: To realize accurate, continuous, and ultrasensitive monitoring of ultraweak pressure stimuli, a desired piezoresistive material should have all of the following structural and material properties: (i) ultralow Young's modulus to significantly reduce the critical stress value that triggers the material deformation; (ii) multilevel or hierarchical structure with ultrahigh densities of variable conductive pathways to allow very large changes in the electrical conductivity of the sensing materials during structural deformation; (iii) excellent mechanical elasticity and robustness to prevent the collapse and disintegration of the material during repeated structure deformations. I agree that a low Young's modulus may help. However, there should be more ways to achieve ultrasensitivity other than what are claimed here. Also, the second and the third points are not convincing.

REPLIES: Thanks for pointing out this problem! We have deleted these sentences and rewritten the introduction part in the revised manuscript (Highlight in Page 3, 4, 5, and 6 in the manuscript).

Comments: Applying a small pressure lower than 0.1 Pa is difficult. I like the method offered by

the authors. They may provide more details on the use of acoustic pressure.

REPLIES: Thanks for the suggestion! A computer-controlled home-made loudspeaker box served as an acoustic source to produce sound or acoustic waves with various frequencies. The sound source was positioned 5 cm over the sensing device and facing right to sensor surface. The precise volume reaching the sensor surface was determined by a decibel meter. The resistance and current changes were measured using a Keithley 2000 digital multimeter. To prevent the environmental noise interference, both the sound source and pressure sensor were installed inside a noise isolation chamber. The sound pressure (P) is defined as $P = P_0 10^{\frac{L_{dB}}{20}}$, where P_0 and L_{dB} denote the reference sound pressure in air (20 μ Pa) and the measured sound pressure level, respectively [Aquacult. Eng. 2007, 37, 125]. We have added these experimental details in the experimental part of the revised manuscript (Highlight on Page 26 in the manuscript).

Comments: There are too much typo and grammatic errors. They should ask a native English speaker to further improve the language.

REPLIES: Thanks for pointing out this problem! A native English speaker had helped us to fully polished the language of the revised manuscript.

REVIEWER COMMENTS

Reviewer #1 (Remarks to the Author):

The authors have carefully addressed all the comments raised by me and I think the manuscript could be accepted in the current form.

Reviewer #2 (Remarks to the Author):

1. Obviously, the authors didn't answer the reviewer 1's questions. For example, the pixel of $5 \times 5 \times 2$ mm is not smaller than pixel $5 \times 5 \times 3$ mm.
2. The author didn't answer comments from Reviewer 2. For example, "what are basic design requirements? Are there any reports or references?"
3. Although the authors have listed the full name of many inexplicable abbreviations in the revised manuscript, but the authors look so much pride and don't realize the problems for reading by many arbitrary abbreviations.
4. The authors still didn't provide direct evidence to verify the structure of cartoon image in Figure 1a, so all the claimed conclusions are not reasonable!
5. I can not agree with the authors' reply for comments "Many references are wrongly cited, such as "...covalently bond with MXene nanosheets via the formation of Ti-O-Si¹²¹" in line 172 (The MXene is first reported in 2011, however, the ref. 21 has already published in 2001.); "PGPDMS covalently cross-links the MXene nanosheets via the formation of Ti-O-Si bonds (Supporting Note 3)²²." (There is no any words about the Ti-O-Si covalent bond between MXene and SiO₂ in ref. 22. It is completely wrong citation.)". The cited ref.21 was published in 2001 but the MXene is first reported in 2011, the claimed citation would mislead the readers obviously!!
6. I can not agree with the authors' reply for comments "The authors have make numerous assumptions such as "Such multilevel nano-channel structure allows the detection of tiny external force by shrinking (or expanding) the space between the nano-channels (meeting requirement (i)). Moreover, numerous new conductive pathways can be created between the neighboring MXene nanosheets during nano-channel shrinking (meeting requirement (ii)), leading to considerable resistance changes inside the cellular walls....." in lines 178-183.". Obviously, the authors didn't carefully answer reviewers' comments and suggestions, just deletes sentences.
7. I can not agree with the authors' reply for comments "The authors have claimed that "ICP-MX-AG are investigated using molecular dynamics (MD) simulations (Figure S4) and Supporting note 2)". However, there is no any direct evidence for verifying the structure of ICP-MX-AG, so what is the sense of simulation? What is the base for using MD simulation?" The authors didn't directly answer reviewers' questions.

8. I can not agree with the authors' reply for comments "The authors claim that ".....expansion of nano-channels in the cellular walls of BBP-MX-AG" in line 256, however, there is no any direct evidence to verify this conclusion. The in-situ characterization only just shows the curving of MXene bundle sheets under loading." The authors didn't carefully answer reviewers' questions. The scale bar in Figure 3 of revised manuscript is 30 nm, but the thickness of a cellular wall changes very little only less 1 nm, which is very difficult to be exactly detected as claimed!!!

9. I totally agree with reviewer 3' comments that "The authors stressed high sensitivity of their sensor. However, high sensitivity at low pressures is not a challenging work.....The authors should do a more comprehensive research on the state of the art of this field." The current manuscript is obviously low level than Nature Communications.

10. I do not agree with the authors' reply for the reviewer 3' comments that "The demonstration of the change in the d-spacings of the MXene layers via in-situ TEM inspection seems amazing (Figure 3). However, I question about the applied stress level in this observation (by using indentation). Since the specimen is significantly deformed, the stress is estimated to be on GPa level, while the sensing test is conducted on 0.001-1 Pa level. The difference is at about 10 orders of magnitude. Obviously, it is not convincing to clarify the sensing mechanism by using the in-situ indentation and TEM observation." The authors did not direct answer reviewer's questions.

In a word, the revised manuscript didn't address all the comments from three reviewers, so it can not achieve high level journal of Nature Communications.

Reviewer #4 (Remarks to the Author):

The manuscript reports a design of bottlebrush polysiloxane cross-linked MXene aerogel with compressible nano-channels inside the cellular walls as active material for ultrasensitive subtle pressure sensing. This is definitely a very interesting work and really pushing the detectability a step forward if all the test were accurate. It's quite impressive that the detect limit of 0.0063 Pa of piezoresistive pressure sensors made by using this material is even below the detect limit of mechanical testing stage. After review the revised manuscript and the replies to former reviewers, I could say this manuscript is basically acceptable. The work proved to be of certain novelty. And the paper was now well-prepared, the experimental results are clear. I believe the manuscript is acceptable for publication after addressing the following questions.

1. According to Fig 4f and Fig S24a, there should be at least 3 stages of the piezoresistive pressure sensors. What is the sensitivity between 0.1 Pa to 200 Pa? Author interpreted the mechanism of second stage as bending or buckling of cellular walls. What happened when the pressure larger than 200 Pa then? Is there any other compression mechanism or is that a result of combined effect of bending/buckling of cellular walls and shrinking of the nanochannels?

2. Please avoid using (c1)(c2)(d1)(d2) for figures. Just use c d e f instead.

3. Author wrote "We first characterized the morphology and structure of BBP-MX-AG and ICP-MX-AG using scanning electron microscopy (SEM), high-resolution transmission electron microscopy

(HRTEM), and X-ray diffraction (XRD).” But there’re no SEM and HRTEM data for ICP-MX-AG in main text and supporting information. Please make sure all the data you want to present are added correctly.

4. There are two terms “MXene-AG” and “MX-AG” appear in the manuscript at the same time. I believe these two nouns are actually refer to the same thing. Please choose only one term and keep consistent in the article.

5. The main difference between BBP-MX-AG and ICP-MX-AG is compressive modulus. In this kind of material system with hierarchical structures, lighter compressive modulus should be directly related to the detection limit that the material can reach. So in this point of view, the authors may consider exploring the relationship between different R1 (substituent of polysiloxane) and compressive modulus.

6. I noticed that in Fig S11. EDS element maps of the scaffold in BBP-MX-AG, the Ti element is elements are distributed everywhere. Based on my understanding of the material model, the Ti element should be on the frame rather than as shown in the Fig S11 now. Could you explain why?

7. It’s quite interesting to learn that authors tried to use a homemade acoustic pressure measurement method to applying pressure lower than 0.1 Pa. More details about calibration of these measurement should be provided. This is not a standard or at least not that common test method when it comes to piezoresistive pressure sensors. Is there anyone else using this method? Please add the references if possible.

REPLY to REVIEWERS
for MANUSCRIPT NCOMMS-21-05897A

Dear Editor:

We appreciate your consideration of our manuscript! We also appreciate the valuable comments and advises from the reviewer#4, which can help to further improve the quality of our manuscript! We are hereby submitting the revised manuscript entitled “*Pushing the Detectability and Sensitivity for Subtle Force Signals to New Limits with a Shrinkable Nanochannel Structured Composite Aerogel*” (NCOMMS-21-05897A) for your consideration for publication in *Nature Communications*. Below are our responses to the reviewer#4’s comments. We have made appropriate revisions to our manuscript according to the comments, and all revisions are highlighted in the revised manuscript.

We do hope the paper after revision would find your approval for publication. Your kind consideration is greatly appreciated. We are looking forward to hearing from you soon.

Sincerely,

Jiajie Liang & Yongsheng Chen

Professor of Materials Science and Engineering

Reply to the reviewer#4:

Comments: The manuscript reports a design of bottlebrush polysiloxane cross-linked MXene aerogel with compressible nano-channels inside the cellular walls as active material for ultrasensitive subtle pressure sensing. This is definitely a very interesting work and really pushing the detectability a step forward if all the test were accurate. It's quite impressive that the detect limit of 0.0063 Pa of piezoresistive pressure sensors made by using this material is even below the detect limit of mechanical testing stage. After review the revised manuscript and the replies to former reviewers, I could say this manuscript is basically acceptable. The work proved to be of certain novelty. And the paper was now well-prepared, the experimental results are clear.

REPLIES: We appreciate the positive comments very much!

Comments: According to Fig 4f and Fig S24a, there should be at least 3 stages of the piezoresistive pressure sensors. What is the sensitivity between 0.1 Pa to 200 Pa? Author interpreted the mechanism of second stage as bending or buckling of cellular walls. What happened when the pressure larger than 200 Pa then? Is there any other compression mechanism or is that a result of combined effect of bending/buckling of cellular walls and shrinking of the nanochannels?

REPLIES: Thanks for the question! The sensing curves in Figure S24a can be divided into three pressure range: 0.025-0.1 Pa, 0.1-150 Pa, and 150-1140 Pa. The corresponding sensitivity under 0.025-0.1 Pa, 0.1-150 Pa, and 150-1140 Pa was 1476.7 kPa^{-1} with a linearity (R^2) of 0.996, 486.2 kPa^{-1} with a linearity of 0.998, and 208.8 kPa^{-1} with a linearity of 0.988, respectively. In the subtle pressure range of 0.025-0.1 Pa, the sensing mechanism is dominated by the shrinking of the nanochannels inside the cellular walls. Under the pressure range of 0.1-150 Pa, the resistance changes are mainly contributed by the bending and buckling of cellular walls. When entering into the pressure range of 150-1140 Pa, corresponding to an abrupt stress increasing regime as can be seen in the compressive stress-strain curves in Figure 2f, continuously decreasing in pore

volume, nearby wall-to-wall contraction and subsequent densification of cells are happened in the sensing aerogel, which majorly contributed to sensing mechanism (*Adv. Mater.*, 2016, 28, 2229; *Nat. Commun.*, 2012, 3, 1241; *Adv. Mater.*, 2013, 25, 2219). Thus, the sensing curve was divided into three stages. We have added these discussions in the main text of the revised manuscript and in the caption of Figure S24 in the revised SI (highlighted in Page 16 and Figure S24).

Comments: Please avoid using (c1)(c2)(d1)(d2) for figures. Just use c d e f instead.

REPLIES: Thanks for pointing out this problem. We have modified the Figure 6 in the revised manuscript (highlighted in Figure 6).

Comments: Author wrote “We first characterized the morphology and structure of BBP-MX-AG and ICP-MX-AG using scanning electron microscopy (SEM), high-resolution transmission electron microscopy (HRTEM), and X-ray diffraction (XRD).” But there’re no SEM and HRTEM data for ICP-MX-AG in main text and supporting information. Please make sure all the data you want to present are added correctly.

REPLIES: Thanks for point out this problem! We have modified the corresponding descriptions in the revised manuscript (highlighted in Page 10 and 11).

Comments: There are two terms “MXene-AG” and “MX-AG” appear in the manuscript at the same time. I believe these two nouns are actually refer to the same thing. Please choose only one term and keep consistent in the article.

REPLIES: Thanks for pointing out this problem! We have replaced all “MXene-AG” with “MX-AG” in the revised manuscript (highlighted in Page 15 and 17).

Comments: The main difference between BBP-MX-AG and ICP-MX-AG is compressive modulus. In this kind of material system with hierarchical structures, lighter compressive modulus should be directly related to the detection limit that the material can reach. So in this point of view, the authors may consider exploring the relationship between different R1 (substituent of polysiloxane) and compressive modulus.

REPLIES: Thanks for the suggestion! In fact, the relationship between different R_1 (substituent of polysiloxane) and compressive modulus is complicated, and more work will be carried out in our future investigations. In the present work, to simply explore the relationship between detection limit and compressive modulus of our sensing aerogels, we have measured the pressure sensing performance of BBP-MX-AG with two different mass densities: $\sim 7 \text{ mg/cm}^2$ and $\sim 10 \text{ mg/cm}^2$. BBP-MX-AG with density of $\sim 7 \text{ mg/cm}^2$ and compressive modulus of 58 Pa exhibited nearly the same pressure detection limit (0.0063 Pa) as the one with a density of $\sim 10 \text{ mg/cm}^2$ and compressive modulus of 140 Pa (Supplementary Figure S28). These results indicated that the compressive modulus may not be the critical factor to determine the pressure sensing limit of our sensing aerogels. This is because the pressure detection limit is mainly determined by the shrinkable nanochannels in the cellular walls of BBP-MX-AG. Both BBP-MX-AG samples have similar multilevel nanochannels in their cellular walls because their initial MXene-to-silane weight ratio and fabrication conditions were the same; however, the sensitivity of BBP-MX-AG with a density of $\sim 7 \text{ mg/cm}^2$ was lower than the one with a density of $\sim 10 \text{ mg/cm}^2$ (Supplementary Figure S28). This is because, at a lower mass density, fewer conductive pathways can be formed during compression, which results in a smaller resistance change under the same applied pressure. Moreover, the detection limit of the pressure sensors assembled from ICP-MX-AG (with density of $\sim 10 \text{ mg/cm}^2$) was only 0.1 Pa (Supplementary Figure S27), despite the presence of shrinkable nanochannels in its cellular walls. MD simulations revealed that the crosslinked PGPTMS network exhibited a much higher compressive modulus than that of bottlebrush-like PGPDMS (Supplementary Figure S9). Thus, a larger force stimulus was required to trigger the deformation of the crosslinked network of PGPTMS in the nanochannels. All these results confirm that the intercalation of a soft polymer with small modulus as the molecular spacer into the nanochannels of multilevel cellular walls plays a critical role in the

fabrication of highly sensitive piezoresistive aerogels. More thorough studies will be carried out in our coming work to investigate this question.

Comments: I noticed that in Fig S11. EDS element maps of the scaffold in BBP-MX-AG, the Ti element is elements are distributed everywhere. Based on my understanding of the material model, the Ti element should be on the frame rather than as shown in the Fig S11 now. Could you explain why?

REPLIES: Thanks for the question! The abnormal phenomenon of the Ti element distribution is mainly caused by the low resolution of the setup. We have replaced the abnormal image with new one as shown in Figure S11 in the revised SI (highlighted in Figure S11).

Comments: It's quite interesting to learn that authors tried to use a homemade acoustic pressure measurement method to applying pressure lower than 0.1 Pa. More details about calibration of these measurement should be provided. This is not a standard or at least not that common test method when it comes to piezoresistive pressure sensors. Is there anyone else using this method? Please add the references if possible.

REPLIES: Thanks for the question! In fact, using a homemade acoustic pressure measurement method to applying pressure has been commonly reported in some previously published works (*ACS Nano* 2021, 15, 1795; *Nat. Commun.*, 2015, 6, 6269). We have cited these two references in the revised manuscript (highlighted in Page 16). As we have described in the Method part of our manuscript, the sound pressure is provided by a homemade loudspeaker box which can produce sound or acoustic waves with various frequencies (similar to Figure S31 in *ACS Nano* 2021, 15, 1795 and Figure S13 in *Nat. Commun.*, 2015, 6, 6269). This sound source was positioned 5 cm over the sensing device and was faced directly at the sensor surface. The output sound wave is edited by “GoldWave” software, and the sound wave template is in “freq” mode under “wave”. The volume of the output acoustic signal was controlled by computer, and the precise volume or sound level reaching the sensor surface was calibrated and determined

by a decibel meter. The resistance and current changes were measured using a Keithley 2000 digital multimeter. The corresponding sound pressure (P) can be calculated by $P = P_0 10^{\left(\frac{L_{dB}}{20}\right)}$, where P_0 and L_{dB} denote the reference sound pressure in air (20 μPa) and the measured sound pressure level, respectively (*Aquacult. Eng.* 2007, 37, 125). Thus, the sound pressure that reaching to the surface of our sensing samples can be precisely controlled by the homemade loudspeaker box. It should be pointed out that to prevent environmental noise interference, the whole measurement setup should be installed inside a noise isolation chamber. We have provided more details about this part in the method section of the revised manuscript (highlighted in Page 26 and 27).

REVIEWERS' COMMENTS

Reviewer #1 (Remarks to the Author):

The reviewer#4 highly recognized the quality of the manuscript and comments raised by the reivewer#4, in my opinion, are well addressed. After carefully checking the revised manuscript, the manuscript could be accepted for publication from my personal point of view.